# Orthogonal proteogenomic analysis identifies the druggable PA2G4-MYC axis in 3q26 AML

Matteo Marchesini[1,2,3], Andrea Gherli[1,2,14], Elisa Simoncini[1,2,14], Lucas Moron Dalla Tor[1,2,14], Anna Montanaro[1,2], Natthakan Thongon[4], Federica Vento[2,5], Chiara Liverani[3], Elisa Cerretani[2,5], Anna D'Antuono[1,2], Luca Pagliaro[1,2,6], Raffaella Zamponi[1,2], Chiara Spadazzi[3], Elena Follini[7], Benedetta Cambò[6], Mariateresa Giaimo[1,2,6], Angela Falco[1], Gabriella Sammarelli[6], Giannalisa Todaro[6], Sabrina Bonomini[6], Valentina Adami[8], Silvano Piazza[8,9], Claudia Corbo[10,11], Bruno Lorusso[1], Federica Mezzasoma[12], Costanza Anna Maria Lagrasta[1], Maria Paola Martelli[12], Roberta La Starza[12], Antonio Cuneo[5,13], Franco Aversa[15], Cristina Mecucci[12], Federico Quaini[1], Simona Colla[4] & Giovanni Roti[1,2,6] ✉

The overexpression of the ecotropic viral integration site-1 gene (*EVI1/MECOM*) marks the most lethal acute myeloid leukemia (AML) subgroup carrying chromosome 3q26 abnormalities. By taking advantage of the intersectionality of high-throughput cell-based and gene expression screens selective and pan-histone deacetylase inhibitors (HDACis) emerge as potent repressors of *EVI1*. To understand the mechanism driving on-target anti-leukemia activity of this compound class, here we dissect the expression dynamics of the bone marrow leukemia cells of patients treated with HDACi and reconstitute the *EVI1* chromatin-associated co-transcriptional complex merging on the role of proliferation-associated 2G4 (PA2G4) protein. *PA2G4* overexpression rescues AML cells from the inhibitory effects of HDACis, while genetic and small molecule inhibition of PA2G4 abrogates EVI1 in 3q26 AML cells, including in patient-derived leukemia xenografts. This study positions PA2G4 at the crosstalk of the EVI1 leukemogenic signal for developing new therapeutics and urges the use of HDACis-based combination therapies in patients with 3q26 AML.

In the last two decades, access to rapidly evolving molecular technologies has dramatically changed the treatment landscape for patients with acute leukemia. Although an increased understanding of the genomic underpinnings of hematopoietic malignancies has led to targeted approaches, especially for mutated kinases, little progress has been made in (i) highly aggressive and rare subgroups, (ii) relapsed and refractory disease, and (iii) leukemia subtypes driven by aberrantly activated transcription factors (TFs).

While TFs have been considered "undruggable" for decades, emerging or improved approaches targeting the modulation of TF activity or expression levels, TF trafficking, and nucleus positioning, or accelerating TF degradation through proteolysis targeting chimeras

---

have repositioned this unique class of drug targets among the most desired cancer therapeutics[1,2]. Several of these approaches showed promising preclinical activity, and a few have been translated into clinical applications or clinical trials, paving the way for transcriptional-based therapy in cancer.

The acute myeloid leukemia (AML) model that carries a 3q26 defect such as the inv(3)(q21q26), t(3;3)(q21;q26), ins(3;3) (q26;q21q26), t(3;8)(q26;q24), and t(3;21)(q26;q22) involving the TF Ecotropic Virus Integration *EVI1* [*MDS and EVI1* complex locus (*MECOM*)][3] in the 3q26 locus exemplifies aggressiveness, with the poorest overall survival (OS) and refractoriness against conventional chemotherapy[4,5].

While 3q26 AMLs have been identified as a distinct subgroup of AML[4,6–9], little has been done to improve their clinical outcome. Most patients with 3q26 abnormalities are misdiagnosed[10] and treated with a cytarabine (ara-C) and anthracycline-based protocol[11], despite early evidence that they do not experience a response to cytotoxic drugs[12,13] or myeloablative allogeneic hematopoietic stem cell (HSC) transplantation[14]. However, targeted repression of *EVI1* based on small molecule, epigenetic, and metabolic approaches impairs leukemia progression in preclinical cancer models, suggesting that EVI1 represents a cancer dependency and a tumor biomarker for high-risk AML[15–17]. While these approaches are promising, efforts to optimize EVI1 suppression for clinical translation are still underway.

Here, we identify a druggable mediator of HDACis response in 3q26 AML and mechanistically demonstrate evidence of targeted suppression of *EVI1* using phenotypic and genome-based in silico small molecule screen approaches and a proteomic analysis of the EVI1 chromatin complex in preclinical-3q26-leukemia-models and leukemia patients.

## Results

### Identification of small molecules modulating EVI1

We performed a phenotypic screen focusing on antiproliferative agents in parallel with a gene-expression-based screen approach to identify molecules that impair cell proliferation by suppressing an *EVI1* signature (Fig. 1A) in AML cells overexpressing EVI1 (EVI1[High]). We screened 5292 drugs or drug-like small molecules, including natural derivative compounds in the human 3q26 EVI1[High] AML cell lines MOLM1 and UCSD/AML1 carrying the chromosome recombinations inv(3)(q21.3q26.2) and t(3;3)(q21.3;q26.2), respectively (Supplementary Data 1). The effect of compound treatment was normally distributed, and the majority of the compounds did not impair cell viability (Supplementary Fig. 1A, B). Plate-specific spatial biases did not affect the experimental screening data (Supplementary Fig. 1C), except for differences due to cell lines. We scored compounds based on their ability to suppress proliferation compared to vehicle controls (ΔPOC). We selected 228 (95th percentile) small molecules to pursue further based on ΔPOC inhibition (Fig. 1B). Among top-scoring hits, with the potential for clinical translation, histone deacetylase inhibitors (HDACis), proteasome modulators, topoisomerase and protein kinase inhibitors reduced the growth of EVI1[High] AML by 90% (Fig. 1C). Of these, 40 small molecules with the highest ΔPOC index (>84%, $Adj.P \leq 2 \times 10^{-33}$) (Supplementary Data 1) were counter-screened in an additional t(3;3)(q21.3;q26.2) EVI1[High] AML cell line (HNT34) and two not-translocated (EVI1[Low]) AML cell lines, NOMO1 (*KRAS*[G13D]) and THP1 (*NRAS*[G12D]); these were selected because they carry mutations in the RAS signaling[18] as in 3q26 AMLs. The HDACis AR-42, belinostat, trichostatin A, and entinostat preferentially affect the proliferation of EVI1[High] AML cells compared to EVI1[Low], suggesting that EVI1 sensitizes 3q26 AML cells to HDACi-mediated cytotoxicity (Supplementary Fig. 1D).

We redefined an *EVI1* transcriptional signature from genome-wide expression profiling of EVI1[High] AML (TF1) transduced with an siRNA targeting *EVI1*[19]. We obtained a set of 1428 modulated genes ($P \leq 0.05$

by the two-sided Student's t-test) and selected the top 100 up- or down-regulated genes, ranked by the signal-to-noise ratio, to define an *EVI1* "On" versus an "Off" state (Supplementary Fig. 2A). We then interrogated publicly available large-scale chemical and genetic perturbation datasets (https://clue.io/cmap)[20] to identify small molecules that mimic *EVI1* transcriptional suppression. HDACis ranked on the top of the hit list were associated with the *EVI1* "Off" status (Fig. 1D and Supplementary Data 1). Most (56.2%) of the HDACis sampled in the CMap libraries scored among the top 250 hits ($P = 0.0006$, Fig. 1E). However, because HDACis are overrepresented in small molecule libraries compared to other classes, we determined whether the size of the compound class ($n = 32$) interfered with our result. Compound classes with similar or higher representation, such as histamine ($n = 52$), glucocorticoid ($n = 47$), or acetylcholine receptor antagonists ($n = 66$), did not show significant enrichment with an *EVI1* "Off" status (Supplementary Fig. 2B). Consistently, a tertiary screen of 4942 small molecules confirmed the potent antiproliferative activity of HDACis compared to non *EVI1* "Off" inducers (Supplementary Fig. 2C) in TF1 cells.

To validate data derived from the analysis of published datasets[19], we downregulated *EVI1* by shRNA in HNT34 cells and detected changes in RNA abundance with more than 2224 transcripts changing from $\log_2$ fold $\geq 2$ (false discovery rate (FDR) $\leq 0.01$) (Supplementary Fig. 2D). The overlap between *EVI1*-regulated gene signatures in HNT34 and *EVI1*-dependent genes in TF1 (Supplementary Fig. 2E) indicates their appropriateness for additional gene-expression-based in silico drug screening approaches. Projection of the HNT34 signature onto the LINCS L1000[21] dataset space demonstrated that transcriptional changes associated with an *EVI1* "Off" status mimic and match HDACis signatures (Fig. 1F, Supplementary Fig. 2F, G and Supplementary Data 1). The suppression of an *EVI1*-transcriptional program due to HDAC perturbation was also replicated in a separate dataset derived from AML cell lines distinguished based on their 3q26 status (Fig. 1F, Supplementary Fig. 2F, G and Supplementary Data 1)[15].

Taken together, these data suggest that HDACis inhibit 3q26 AML cell proliferation and modulate *EVI1*, supporting their investigation as a targeted approach in 3q26 AMLs.

### HDACis suppress 3q26 AML growth

To set conditions for validating our results, we confirmed the expression of EVI1 and ΔEVI1 protein isoforms[22] in the 3q26 EVI1[High] AML cell lines (Fig. 2A, B) and demonstrated the dependency of these cells on EVI1 expression (Fig. 2C–E, Supplementary Fig. 3A–C). Suppression of *EVI1* abrogates the growth and the clonogenic capacity of EVI1[High] AML (Fig. 2E, F and Supplementary Fig. 3C, D).

We then selected a panel of molecules with pan (AR-42, belinostat) or selective (entinostat, class 1/2) HDAC inhibitory activity based on their potential in clinical translation in high-risk-AML. AR-42 (ΔPOC = 99.54; $Adj.P = 1.51 \times 10^{-34}$) has been investigated in clinical trials for the treatment of hematologic malignancies (e.g. NCT01129193 and NCT02569320); belinostat (also known as PXD101, ΔPOC = 98.14; $Adj.P = 1.51 \times 10^{-34}$) has been approved for cutaneous of T-cell lymphoma[23]; and entinostat (ΔPOC = 87.43; $Adj.P = 7.55 \times 10^{-34}$) has been tested in myeloid malignancies, irrespective of the genetic status (e.g. NCT00313586 and NCT01159301)[24]. All molecules displayed half-maximal inhibitory concentrations (IC$_{50}$) within the low micromolar range: AR-42 from 0.221 μM to 0.438 μM (0.329 ± 0.153), belinostat from 0.379 μM to 0.951 μM (0.665 ± 0.404), and entinostat from 0.812 μM to 1.686 μM (1.249 ± 0.618) (Fig. 2G). Treatment with HDACis also induced a dose-dependent increase in apoptosis (Fig. 2H, I and Supplementary Fig. 3E, F), and suppressed blast colony formation (Fig. 2J, K).

### EVI1 sensitizes 3q26 AML to HDAC-mediated suppression

To test whether the effects of HDACis were due to EVI1 suppression, we treated EVI1[High] AML cell lines with AR-42, belinostat, and entinostat

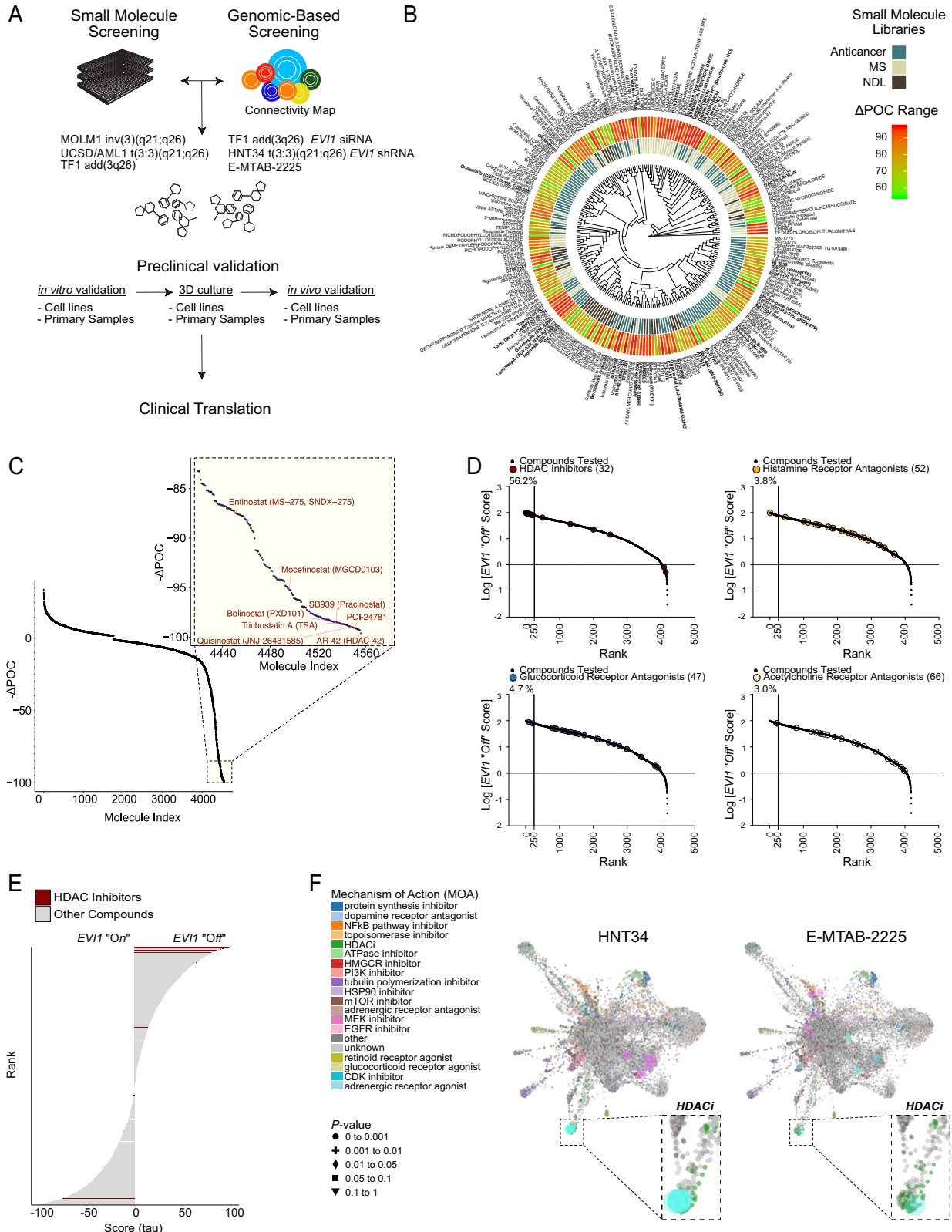

and observed a loss of EVI1 both at transcriptional and protein level. In contrast, cleaved caspase 3 was regulated in the opposite direction, consistent with the apoptosis data presented above (Fig. 3A, B and Supplementary Fig. 4A).

To further assess whether EVI1 sensitizes AML cells to HDAC inhibition, we leveraged an EVI1-inducible leukemia model[25]. U937T cells that conditionally expressed the full-length isoform of *EVI1*

under the control of the tetracycline promoter were grown in the presence (*EVI1 "Off"*) or absence of tetracycline (*EVI1 "On"*). EVI1 expression was observed 24 hr after tetracycline removal (Fig. 3C and Supplementary Fig. 4B). U937T (E10 clone) were then treated with AR-42, belinostat, and entinostat in a 2-fold dilution series for three additional days. Despite a growth difference between U937T *"On"* or *"Off"* (Supplementary Fig. 4C), cells grown in the absence of

**Fig. 1 | Identification of *EVI1* modulators from the intersection of phenotypic and genome-based approaches. A** Identification of *EVI1* "*Off*" modulators by the integration of phenotypic and in silico gene-expression-based screen approaches. **B** Circos plot summarizing primary small molecule screening in EVI1[High] AML cell lines. The heatmap at the external circle shows the normalized percentage of cell death (ΔPOC) for the 95th percentile for the compounds tested. The middle circos shows the compound libraries: i) Spectrum Collection (*n* = 1914); ii) the anti-cancer compound library (*n* = 343); and iii) the NDL-3000 library (*n* = 3035). The inner plot shows a hierarchical cluster of top candidates' scalar fingerprints (SMILE), based on the Tanimoto similarity metric. **C** Dot plot showing the effects of the compound tested (*n* = 5292), ranked based on ΔPOC. The yellow area indicates the 95th percentile. HDAC inhibitors (HDACis) are labeled in red. **D** Rank-score log-scale plots for HDACis (*n* = 32), histamine (*n* = 52), glucocorticoid (*n* = 47), and acetylcholine

receptor antagonists (*n* = 66). In black, the number of perturbagenes inferred in the connectivity map of which 2836 are small molecules. Compounds of interest are color-coded and expressed as a percentage relative to the entire library. **E** Bar plot showing the HDACis (in red) ranking in the ConnectivityMap[20] dataset. The tau score represents the level of similarity to the *EVI1* "*On*" or "*Off*" status. **F** L1000FWD fireworks visualization of drug-induced signatures mimicking and reversing the differential gene expression signature generated from *EVI1*-silenced HNT34 (left) and from EVI1[High] (*n* = 5) or EVI1[Low] (*n* = 2) AML cell lines contained in the E-MTAB-2225 dataset (right). Aquamarine circles indicate small molecules causing a reverse signature *EVI1* "*Off*". Light violet circles indicate small molecules mimicking a signature *EVI1* "*On*". HDACis drug-induced signatures are indicated in green overlapping with aquamarine circles. See also Supplementary Fig. 1-2 and Supplementary Data 1.

tetracycline, with the highest EVI1 expression, were more sensitive to the effects of HDACis (Fig. 3D). EVI1[High] and EVI1[Low] U937T were equally sensitive to ara-C, suggesting that the expression of EVI1 preferentially enhances the HDACi-induced inhibition of cell viability compared to chemotherapy (Fig. 3D and Supplementary Fig. 4D). Similar results were obtained in a constitutive model of *EVI1* expression in HL-60 cells (Fig. 3E, F and Supplementary Fig. 4E). We then sorted MOLM1 cells by size, distinguishing EVI1-positive cells with large nuclei and prominent nucleoli, and EVI1-negative cells, smaller in size with weak or no EVI1 protein expression (Figs. 2B and 3G, H). In this isogenic model HDACi were more active in MOLM1 EVI1[High] cells compared to MOLM1 EVI1[Low] (Fig. 3I). Consistently, EVI1[High] (*n* = 5) and EVI1[Low] (*n* = 7) AML cell lines were equally sensitive to other chemotherapy agents but not to AR-42, belinostat, and entinostat (Fig. 3J).

In addition, the interaction of 3q26 AML cells with a collagen substrate in a three-dimensional (3D) porous and cell-permeant culture system[26] (Supplementary Fig. 4F) used to mimic the bone marrow microenvironment did not rescue the cytotoxic effect of HDACis. Here, EVI1[High] cells aggregate in niche-like structures similar to AML clusters in the bone marrow stroma, retain EVI1 expression, and grow with a doubling time of ~6 days (Supplementary Fig. 4G–I). EVI1[High] cells cultured in biomimetic 3D collagen scaffolds develop resistance to ara-C but not to HDACis, suggesting that cell-extrinsic mechanisms of acquired resistance to chemotherapy agents can be overcome by targeting EVI1 in 3q26 AML (Supplementary Fig. 4J).

Collectively, these data demonstrate that EVI1 expression in 3q26 AML sensitizes cells to HDAC inhibition, providing a rationale for translating these molecules in clinical practice.

## HDACis suppress EVI1 in patients with 3q26 AML

Since January 2017, we have admitted five patients to our hospital affected by AML carrying 3q26 abnormalities and collected six additional samples from different institutions (PR#001-009 and PR#023-024). Four out eleven patients had atypical 3q26 rearrangements (PR#001, PR#004, PR#005, and PR#024) (Supplementary Fig. 5A and Supplementary Data 2). Sequencing and cytogenetic analyses confirmed a high frequency of RAS mutations[27] (63.6% mutated in *KRAS* or *NRAS*) and monosomy of chromosome 7 (63.6%) compared to non-rearranged cases (PR#010-022 and PR#025-039) (Supplementary Fig. 5B and Supplementary Data 2). All patients carrying 3q26 abnormalities presented severe clinical conditions and experienced disease relapse after multiple lines of chemotherapy or HSCT (Supplementary Data 2). In primary AML blasts, suitable for ex vivo studies, AR-42, belinostat, and entinostat inhibited cell viability in 2D and 3D models (Fig. 4A, B and Supplementary Fig. 6A). 3q26 AML cells collected from clinical samples (*n* = 9) were more sensitive to HDAC suppression than were other AML subtypes (*n* = 28) and equally responsive to chemotherapy agents in vitro (Fig. 4C). For samples with adequate cells for additional tests, EVI1 expression was almost undetectable after 24 hr of HDACis treatment, consistent with the disappearance from the nucleus, and inversely correlated with the

increase of cleaved caspase 3 (Fig. 4D–F). Importantly, ara-C did not cause significant changes in EVI1 nuclear protein abundance (Fig. 4E, F and Supplementary Fig. 6B).

Three patients (PR#001, PR#002, and PR#003) were enrolled in a compassionate use program that tested azacitidine at 50 mg/m$^2$ (days 1–10) and entinostat 4 mg/m$^2$ (days 3 and 10), every 28 days. This schedule was approved in the NCT00101179 clinical trial for AML or myelodysplastic syndromes[28]. Two patients received standard support or chemotherapy (PR#004 and PR#005); the remaining were treated in a different hospital. The mean survival duration of patients receiving an entinostat-based regimen was 20.3 months without significant organ toxicity compared to 4.5 months for those receiving standard of care (Supplementary Fig. 6C, D and Supplementary Data 2).

Leveraging previous studies describing the pharmacokinetics properties of entinostat[29], we also demonstrated that EVI1 protein levels markedly decrease in circulating AML blasts following 4 mg/m$^2$ of entinostat compared to ara-C both in treated patients (Fig. 4G) or in the matched 3q26 PDLX models (Supplementary Fig. 6E).

In summary, these data support the idea of testing HDACi-based combination therapy in patients with 3q26 AML in randomized clinical trials and suggest the use of EVI1 as a potential biomarker of response to epigenetic treatment or resistance to standard of care (Fig. 4H) in patients with EVI1-dependent leukemia.

## EVI1 inhibition modulates the Myc transcriptional program

We then hypothesized that the preferential activity of HDACis in 3q26 AML may be related to the impaired function of the EVI1 co-transcriptional complex. We profiled EVI1[High] AML cell lines after 16 hr of treatment with AR-42 (0.5 μM and 1.0 μM, respectively) and entinostat (2 μM and 4 μM, respectively) by mRNA and ChIP-Seq. Consistent with the role of HDAC in chromatin-dependent transcriptional regulation, there was a preponderance of up-regulated genes (1309) versus down-regulated genes (107) (Supplementary Fig. 7A and Supplementary Data 3). To assess the effects of HDACis on the transcriptional programs regulated by *EVI1*, we interrogated our data with four well-validated gene signatures for statistically significant enrichment by single-sample gene set enrichment analysis (ssGSEA)[30]. We incorporated the following databases: Molecular Signatures Database (MSigDB) of 50 gene sets, Elsevier of 1721 gene sets, Kyoto Encyclopedia of Genes and Genomes (KEGG) of 308 gene sets, and BioPlanet datasets of 1510 gene sets (Supplementary Data 3). In each database, we combined EVI1[High] datasets derived from the analysis of *EVI1* upregulated signatures in primary myeloid leukemias cells carrying 3q26 abnormalities (GSE14468, GSE134589)[31,32]. Next, we determined whether any of the gene sets in our selected collection were significantly enriched in EVI1[High] AML treated with DMSO (control) compared to those treated with HDACis. Data from the four databases revealed that treated samples were strongly correlated with signatures associated with the modulation of *MYC* oncogene (Fig. 5A, Supplementary Fig. 7B and Supplementary Data 3).

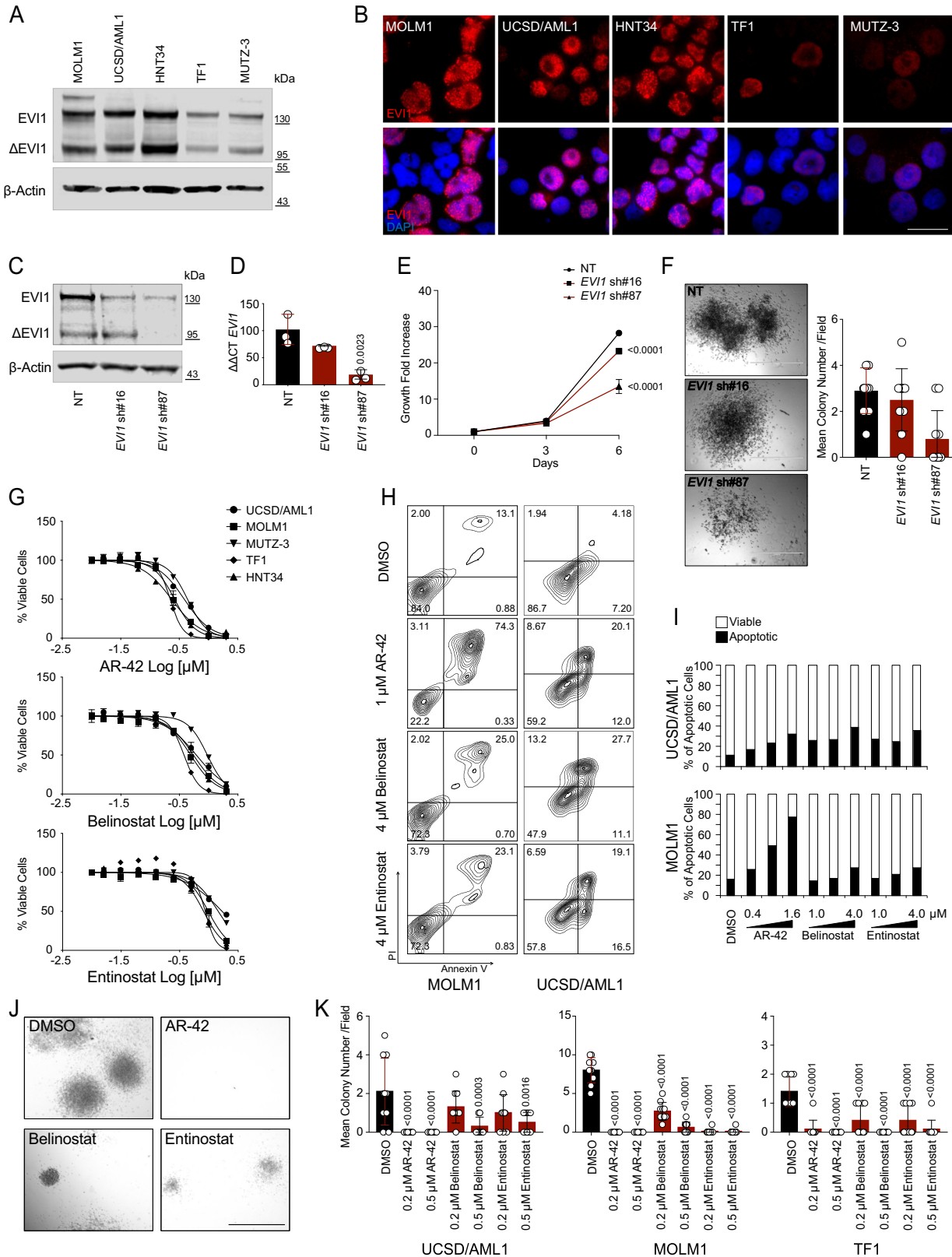

We then set up a longitudinal study and profiled bone marrow cells of PR#002 by single-cell RNA sequencing (scRNASeq) before and after four cycles of azacitidine and entinostat (Supplementary Data 3). A cluster dimensional analysis revealed 18 groups of transcriptionally distinct cell populations, among which 15 were represented in both samples (Fig. 5B and Supplementary Data 3). After treatment, we observed a reduction of the more undifferentiated myeloid

compartment and a stimulatory effect, as seen before with HDACis[33], on the immune system cells (Supplementary Fig. 7C).

Next, we computed a score based on the combined expression of the patients' leukemic markers (*CD34, KIT, CD33, ANPEP, CD38, CD7, NCAM1, CD4, CD19, DNTT, ITGB3, HLA-DRA, HLA-DRB1*, and *HLA-DRB5*) using UCell[34] and identified the leukemic population (LP) in the PR#002 clone as the group of cells with score >0.2. This

**Fig. 2 | Validation of HDACis as *EVI1* modulators. A** EVI1 and ΔEVI1 expression in EVI1^High AML cell lines (*n* = 3 biological replicates). **B** Nuclear localization expression of EVI1 (in red) detected by immunofluorescence (IF) in EVI1^High AML cell lines (*n* = 3 biological replicates). Cell nuclei were stained with DAPI (blue), scale bar: 10 μm. **C** EVI1 and ΔEVI1 expression in TF1 cells 6 days after shRNA transduction. **D** Percentage of *EVI1* mRNA relative to the control gene *RPL13A* (ΔΔCT) in TF1 cells 6 days after shRNA transduction. **E** Effect of *EVI1* loss in TF1 cells at three or six days after shRNA selection on cell proliferation. **F** Effect of shRNA-directed *EVI1* loss in the ability of TF1 to form colonies in methylcellulose compared to NT (*n* = 2 biological replicates). Scale bar: 1000 μm. The histogram on the right shows the mean number of colonies per field 20 days after plating cells. NT = non-targeting, sh#16 and sh#87 = shRNA directed against *EVI1* (**C**–**F**). **G** Effect of AR-42, belinostat, and entinostat on cell viability in EVI1^High cell lines following 72 hours (hr) of treatment.

**H** Annexin V/propidium iodide (PI) staining after 72 hr of HDACis treatment. Events ≥ 20,000. **I** Apoptotic fold increase, expressed as a percentage of annexin V-positive cells relative to vehicle control. **J** Effect of HDACis on the ability of 3q26 AML cell lines to form colonies in methylcellulose. Representative images of UCSD/AML1 treated with vehicle (DMSO), 0.5 μM AR-42, 0.5 μM belinostat, and 0.5 μM entinostat (*n* = 3 biological replicates). Scale bar: 1000 μm. **K** The histogram shows the mean number of colonies per field 20 days after plating cells (*n* = 3 biological replicates). Statistical significance among groups was determined by one-way (**D**, **K**) or two-way (**E**) ANOVA using Tukey's correction for multiple comparison testing. Data are presented as mean ± standard deviation (SD) in **D** (*n* = 3), **E** (*n* = 4), **F** (*n* = 10 fields per condition), **G** (*n* = 2), and **K** (*n* = 10 fields per condition). Source data are provided as a Source Data file. See also Supplementary Fig. 3.

population accounts for 29.5% of the sample and is predominantly composed of hematopoietic stem cells (HSCs) (61.4%) and lympho-myeloid primed progenitor (LMPP) (31.6%) cell types. After therapy, LP decreased to 3.1% while maintaining a similar proportion of the cluster distribution (70.7% HSCs and 20.2% LMPP) if compared with the sample before treatment (Fig. 5C, Supplementary Fig. 7D and Supplementary Data 3).

In HSCs, LMPP, or LP cells, highly expressed genes are correlated with Myc-associated gene sets, suggesting an inhibitory effect on Myc signaling after azacitidine and entinostat treatment in human 3q26 AML (Fig. 5D, Supplementary Fig. 7E and Supplementary Data 3). Consistently, we observed a reduction in MYC expression in the leukemic cells in the bone marrow of PR#002 after two cycles of azacitidine and entinostat (Fig. 5E). Perhaps it was not surprising that genetic or HDACi-mediated suppression of *EVI1* led to a decrement of MYC in ex vivo EVI1^High AML patient blasts and cell lines (Fig. 5F, Supplementary Fig. 7F, G). This result is consistent with ChIP-Seq findings that reveal a biding of *EVI1* at the *MYC* promoter locus markerd by acetylated histone H3K27 (H3K27Ac). This pattern is significantly diminished following treatment with HDACis (Supplementary Fig. 7H).

Next, we asked whether azacitidine contributed to the clinical response seen in 3q26 AML patients and whether azacitidine played a part in the observed phenotypes. Then, we recapitulated our clinical protocol testing azacitidine and entinostat in a 3q26 PDLX model. We showed that entinostat controls proliferation, *EVI1* modulation and a Myc signaling of PDLX_PR#008 LP (Fig. 5G-I and Supplementary Fig. 7I) with minimal contribution from azacitidine (Fig. 5J–L).

Finally, to exclude a direct effect of HDACis on MYC as previously reported for other cancers[35–37] we compared the effects of AR-42, belinostat and entinostat in AML MYC^High versus MYC^Low (Supplementary Fig. 8A, B) or by overexpressing *MYC* in the U937T inducible model (Supplementary Fig. 8C, D). Differently from *EVI1*, in these models *MYC* did not render cells more sensitive to HDAC inhibition whereas, as expected, it exhibited this effect for JQ-1, a potent repressor of *MYC* superenhancers (Supplementary Fig. 8E, F)[38]. While we cannot exclude an effect of azacitidine on the leukemia response in humans, these results suggest that an HDAC epigenetic-based therapy limits Myc pathways in 3q26 AML and that azacitidine-based protocols may not be as effective in 3q26 AML as they are in other AML subgroups.

## PA2G4 bridges EVI1 with Myc signaling

To prioritize any significant pathway scored by ssGSEA, we dissected the EVI1 chromatin-associated complex by rapid immunoprecipitation mass spectrometry (MS) of endogenous proteins (RIME)[39]. We performed these experiments in UCSD/AML1 and HNT34 cell lines and cells derived from the PDLX_PR#003 model. This model retained the parental t(3;3)(q21.3;q26.2) and EVI1 expression in the tumor mass (Supplementary Fig. 9A, B). We identified 107 EVI1-interacting proteins in HNT34 cells, 117 in UCSD/AML1, and 155 in the PDLX model sample

(Fig. 6A). Sixty-nine were common in all samples (Supplementary Fig. 9C and Supplementary Data 4). Among EVI1 protein partners, we found that translation-related proteins were represented prominently, as well as translational initiating factors (1), ribosomal proteins (47) and RNA binding factors (2), transcription factors (2), histones (2), and DNA and RNA processing proteins (4). A small group of proteins with heterogeneous functions (CENPV, RPN1, FBL, SND1, PARP1, PA2G4, PABPC1, RACK1, MTDH, and ESYT1) was also recovered. In addition, EVI1 partners identified by stable isotope–labeling amino acids in cell culture-based quantitative proteomics, such as PARP1, XRCC5, RPS19, and H2AZ[40] (Supplementary Data 4), were also identified by our approach.

We intersected the 69 hits with the enriched genes scored by ssGSEA. Eleven of the 69 RIME targets were represented in the *MYC Target V1 pathway*. In all other instances, the intersection resulted in two or fewer hits, suggesting that EVI1 and Myc signaling share co-regulators in their transcriptional machinery (Supplementary Fig. 9D and Supplementary Data 4). Ten out eleven these proteins were confirmed by repating the analysis of MS RIME data with a label-free quantification (LFQ) approach (Supplementary Fig. 9E and Supplementary Data 4). A difference in *EVI1*-dependent transcriptional regulation was demonstrated with HDACis in only 3 of 11 common hits: Fibrillarin (*FBL*: $\log_2$ fold change = −0.74, *Adj.P* = 0.017), nucleolar RNA helicase 2 (*DDX21*: $\log_2$ fold change = −0.69, *Adj.P* = 0.006), and, proliferation-associated 2G4 (*PA2G4*: $\log_2$ fold change = −1.45, *Adj.P* = $1.34 \times 10^{-06}$) (Fig. 6B, C and Supplementary Data 3) that was retained for validation since, in our model, PA2G4 interacts with EVI1 (Fig. 6A and Supplementary Fig. 9F, G) and is transcriptionally repressed by HDACis treatment (Supplementary Fig. 9H).

PA2G4 is involved in protein translation, cell cycle progression[41–43], and a cancer-druggable feed-forward transcriptional regulator of *MYCN* in neuroblastoma[44]. Similarly, *PA2G4* is positively correlated with *MYC* expression in AML primary datasets (GSE14468, n = 524 and GSE134589, n = 672; Supplementary Fig. 9I)[45,46] and interestingly, both *EVI1* and *MYC* occupy the promoter region upstream *PA2G4* transcription site (Supplementary Fig. 9J), suggesting that these genes are required for *PA2G4* control in 3q26 leukemia (Fig. 6B).

We next addressed whether the effect of HDACis on *EVI1* was at least partly related to PA2G4 and showed that *PA2G4* knockout abrogates EVI1 and consequently MYC protein (Fig. 6D); it then prevents the growth and formation of colonies in 3q26 AML (Supplementary Fig. 10A, B). Conversely, the overexpression of *PA2G4* increased EVI1 and MYC protein levels (Fig. 6E and Supplementary Fig. 10C) and partially rescued AML cells from the apoptotic effect of HDAC inhibition (Fig. 6F), suggesting that PA2G4 is a putative mediator of HDAC inhibition.

Because genetic loss of *PA2G4* occurs in the absence of transcriptional regulation of *EVI1* or *MYC* (Fig. 6G), we speculated that PA2G4 protects EVI1 from proteasome degradation, as seen for MYCN in neuroblastoma[44]. Consistently, MG132 treatment partially rescues

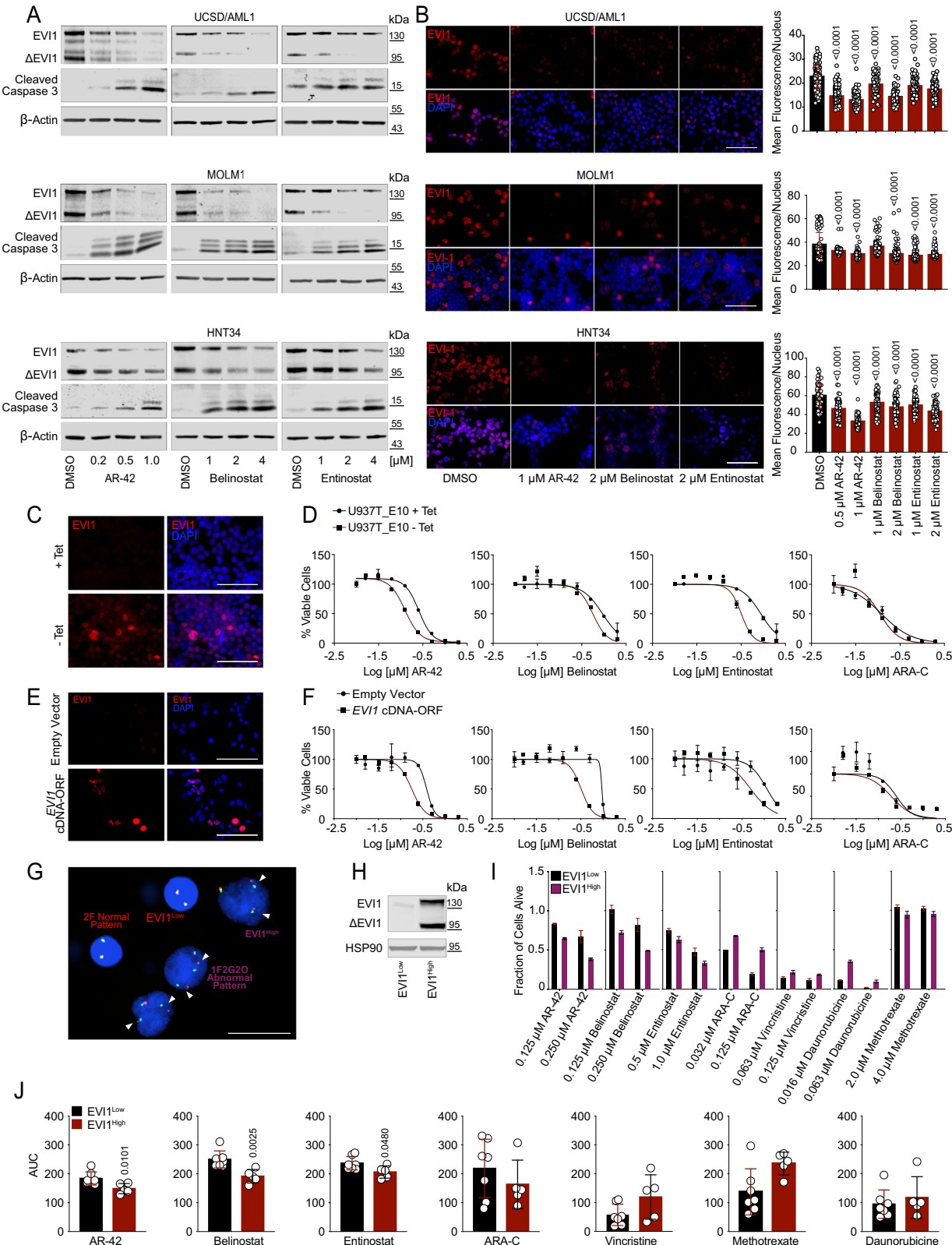

the EVI1 level in 3q26 AML cell lines (Fig. 6H and Supplementary Fig. 10D), suggesting that PA2G4 acts as a scaffolding protein for *EVI1* transcriptional complex.

Taken together, our data suggest that PA2G4 bridges EVI1 to MYC and supports the disruption of this protein to diminish their respective oncogenic signals in this leukemia subtype.

### Selective PA2G4 inhibitor WS6 depletes EVI1 and MYC signaling and blocks 3q26 AML in vivo

The selective PA2G4 inhibitor, WS6, was recently identified on high-throughput screening for inducers of β pancreatic islet cells[47]. WS6 is a dyarilurea compound that interferes with PA2G4-MYCN binding[44]. Given the homology of *MYC* to *MYCN*[48], we tested this molecule with

**Fig. 3 | EVI1 sensitizes AML to HDACis. A** EVI1, ΔEVI1, and cleaved caspase 3 expression in 3q26 EVI1[High] cell lines after 24 hr of treatment with DMSO or HDACis (n = 2 biological replicates). **B** Representative IF images of EVI1 (red) in 3q26 EVI1[High] cell lines (left, n = 3 biological replicates). Cell nuclei were stained with DAPI (blue), scale bar: 100 μm. Right: fluorescence intensity of nuclear EVI1 content in 3q26 EVI1[High] AML cell lines after 24 hr of treatment. **C** IF images of U937T_E10 cells cultured in the presence (EVI1[Low], top panel) or absence (EVI1[High], bottom panel) of tetracycline, incubated with an anti-EVI1 (red) antibody (n = 2 biological replicates). Nuclei in blue (DAPI). Scale bar: 100 μm. **D** Effect of HDACis and ara-C on cell viability after 72 hr of treatment in U937T_E10 cell cultured in the presence (EVI1[Low], black line) or absence (EVI1[High], red line) of tetracycline. **E** Representative IF showing HL-60 cells transduced with an empty (top panels) or an ORF-*EVI1* cDNA (bottom panels) vector and stained with an anti-EVI1 antibody (in red, n = 2 biological replicates). Nuclei in blue (DAPI). Scale bar: 100 μm. **F** Effect of HDACis and ara-C on cell viability after 72 h of treatment in HL-60 ± ORF-*EVI1* cDNA. **G** Abnormal 3q26 pattern on fluorescence in situ hybridization (FISH) in MOLM1 sorted by cell size. The break-apart hybridization pattern 1F1G1O (one fusion and two separated

signals, one green and one orange) indicates the break and split of the *EVI1* locus. The abnormal pattern was observed in cells with large nuclei. At least 100 nuclei/cells were analyzed (n = 2 biological replicates). Scale bar: 20 μm. **H** Expression of EVI1 in MOLM1 sorted based on cell size. Large cells express EVI1 compared to small cells. **I** Effects of HDACis and chemotherapy treatment on viability in MOLM1 EVI1[High] and MOLM1 EVI1[Low] after 72 h of drug exposure. **J** Effect of AR-42, belinostat, entinostat, ara-C, vincristine, methotrexate, and daunorubicin on EVI1[High] (MOLM1, UCSD/AML1, HNT34, TF1, and MUTZ-3) and EVI1[Low] (NOMO1, MOLM13, OCI/AML1, OCI/AML2, GDM1, SKM1, and IMS-M2) AML cell lines calculated using the area under the curve (AUC) model of the log-transformed dose-response data. A lower AUC corresponds to greater sensitivity. Statistical significance was determined by two-tailed non-parametric t-test (Mann-Whitney) (**J**), one-way ANOVA with Tukey's correction for multiple comparison testing (**B**). Data are presented as mean ± SD in **B** (UCSD/AML1 n = 828, MOLM1 n = 1028, HNT34 n = 571), **D** (n = 2), **F** (n = 2), **I** (n = 2) and **J** (EVI1[High] n = 5, EVI1[Low] n = 7). Source data are provided as a Source Data file. See also Supplementary Fig. 4.

the functional assays described above. WS6 inhibited cell proliferation greatly in AML EVI1[High] vs. AML EVI1[Low] (Figures 7A–C and Supplementary Fig. 11A) and potentiate the effect of HDACis (Supplementary Fig. 11B). To exclude the WS6 off-target effect, we demonstrated that WS6 lacks activity against HDAC in biological assays, as shown by the comparative quantification of H3K27 acetylation and β-tubulin in EVI1[High]-treated AML cell lines (Supplementary Fig. 11C). WS6 markedly decreased MYC and EVI1 in EVI1[High] primary blasts and EVI1[High] AML cell lines (Fig. 7D and Supplementary Fig. 11D) and consequently their occupancy at the *PA2G4* promoter (Supplementary Fig. 9J). Similarly to HDACis, WS6 diminished EVI1 expression in circulating AML cells in 3q26 PDLX models in vivo (Fig. 7E). WS6 is a tool compound that has not yet been optimized for continuous systemic delivery; thus, we tested the tumor-suppressing capacity of this molecule by treating two 3q26 AML PDLXs (PDLX_PR#003 and PDLX_PR#008) models at 25 mg/kg for 5 days/week for 15 days. We demonstrated that IP administration of WS6 inhibited LP progression, compared to vehicle-treated control, traced by hCD45+ (Fig. 7F and Supplementary Fig. 11E) or scRNASeq quantification (Fig. 7G and Supplementary Fig. 11F). No major toxicities were seen in treated animals. In addition, EVI1 and MYC protein levels were diminished in WS6-treated tumors compared to vehicle-treated control animals (Fig. 7H, Supplementary Fig. 11G), linking growth inhibition (Ki67) to the suppression of EVI1-MYC.

Taken together, these results demonstrate that PA2G4 is a druggable mediator of EVI1 complex and suggest that PA2G4 inhibitors should be selectively optimized for clinical applications.

## Discussion

Despite recent advancements in the treatment of low- and intermediate-risk AML by targeting mutationally activated kinases[49–51] and newer initial treatment options for a subset of patients[52,53], high-risk AML with recurrent genetic abnormalities, such as inv(3)(q21.3q26.2) and t(3;3)(q21.3;q26.2) remains a significant clinical challenge. In these patient subgroups, induction[54,55] and maintenance cytotoxic chemotherapy have consistently failed to show a benefit[56]. Patients with 3q26 AML, for example, eventually experience relapse, even after allogeneic hematopoietic cell transplantation with curative intent[57,58].

In this case, the development of a small molecule that can block the activity of EVI1 would be an attractive approach[59–61]. Given this temporary absence, an alternative is to focus on unbiased strategies, searching for modulators in one or more leukemogenic steps in the metabolic, transcriptional, or epigenetic pathways that are aberrantly activated by EVI1 in 3q26 AML[15,17]. Closer to clinical translation, the PARP inhibitor emerged as a transcriptional repressor of EVI1 by decreasing the interaction frequency between the *GATA2* distal hematopoietic enhancer (G2DHE) and the *EVI1* promoter[62]. However,

the risk of secondary myelodysplastic syndrome and AML in cancer patients treated with PARP inhibitors such as olaparib and veliparib[63] raises concerns about their use in myeloid leukemia and hematological malignancies more broadly.

Our strategy was meant to identify clinically ready solutions to modulate EVI1 in 3q26 AML. Among potential targets, we decided to validate structurally different pan-HDACis and more selective molecules in various stages of clinical development and subsequently identified a potential mediator of this response.

The functional interaction of EVI1 with the HDAC complexes was initially postulated in heterologous cells transfected with different *EVI1* cDNAs isoforms[64–66] or by quantitative proteomics combined with yeast two-hybrid screens and subsequently validated by coimmunoprecipitation experiments with endogenous EVI1 in cancer cell lines[40]. While collectively, these studies suggest that EVI1 mainly associates and cooperates with HDAC1[40,65,66], HDAC2[40] (class I), or HDAC4 (class II)[67], the number of studies validating EVI1-HDAC co-transcriptional complex in human 3q26 AML and cancer models is surprisingly low.

Nevertheless, our results for EVI1 expression upon HDACis treatment place this compound class in a fascinating scenario. While HDACis have been shown to induce histone hyperacetylation, apoptosis, and anti-proliferative activities against a wide range of rearranged leukemia cell lines or mouse models preclinically, no pharmacodynamic demonstrations exist of their target engagement in AML clinical trials. Further research will be crucial to uncover their potential as therapeutic agents in 3q26 AML, particularly with the next generation of selective HDACis.

*MYC* is broadly altered in AMLs[68] and it is an independent prognostic factor in high-risk AMLs, especially those associated with myelodysplasia-related changes (AML-MRC)[69–71]. Ottema and colleagues recently found that the translocation of the *MYC* super-enhancer (SE) leads to *EVI1* overexpression in t(3;8)(q26;q24) AML[72]. This data suggests that therapeutic approaches that interfere with TFs and co-activators docking to this prototypical location (SE-EVI1 promoter) negatively affect EVI1 expression in t(3;8)(q26;q24) or other 3q26 rearranged cases. However, the identification of EVI1 interactome in AML is still lacking. Previous studies identified potential EVI1 interactors in different tumor types[40,73–76] but, given the heterogeneity of these models, the number of genes overlapping across these approaches is low.

While we confirmed the co-immunoprecipitation of previously reported putative EVI1 co-regulators, such as PARP1, XRCC5, RPS19, and H2AZ[40], we did not observe enrichment for HDACs or MYC by RIME. The interaction with the HDAC machinery, and hence the response to therapy, might be indirect and mediated by different co-regulatory proteins. We showed that PA2G4 contributes to linking HDAC response to EVI1 loss and, in turn, to MYC suppression.

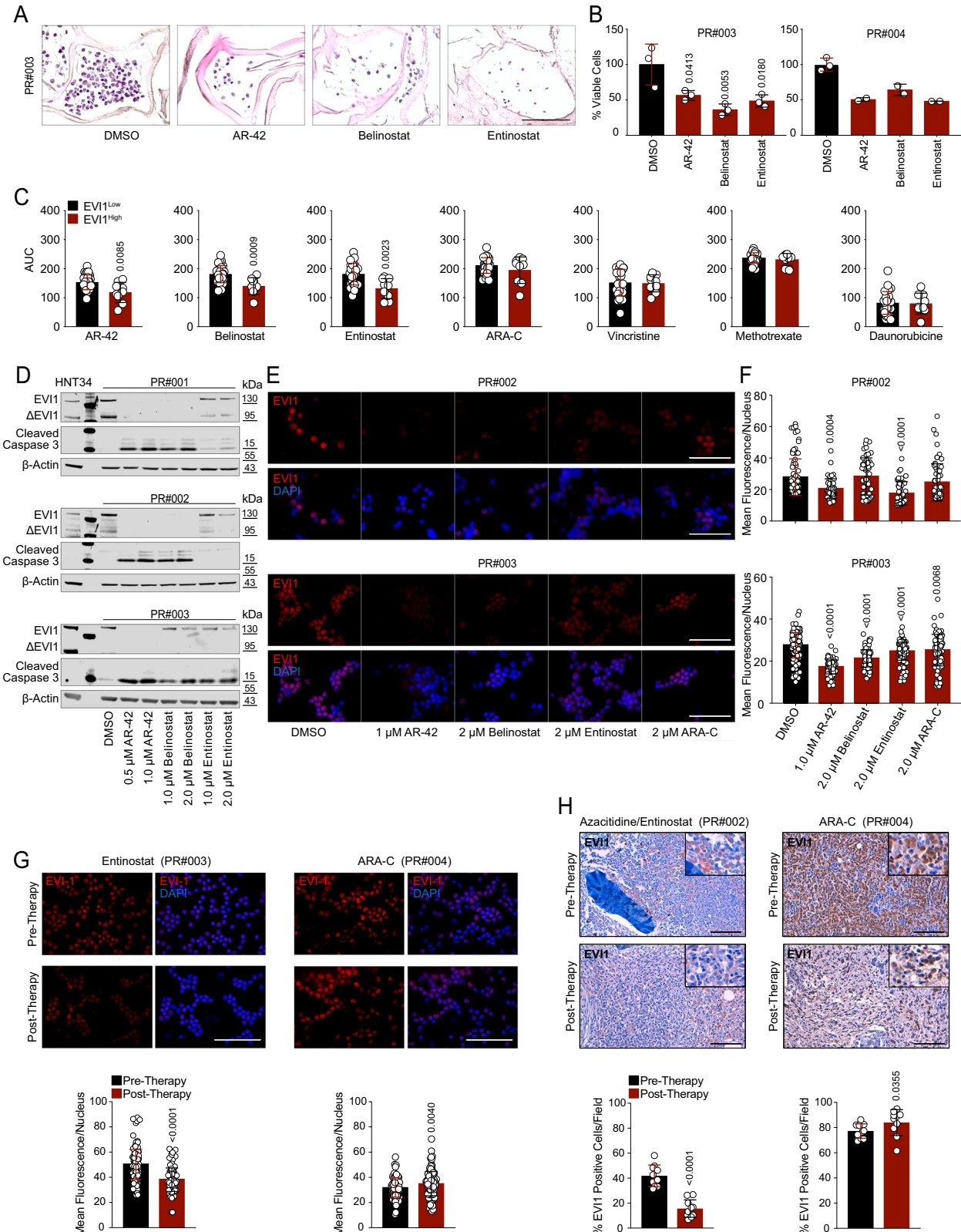

*PA2G4* accounts for 10 exons and encodes two splice *PA2G4* (also known as ErbB3-binding protein 1, EBP1) variants, p48 and p42. The difference, 54 amino acids in the N-terminus, between the long and short isoforms directs unique function, oncogenic or tumor suppressive[77], and association with different binding partners and their regulation[78]. p48 PA2G4 is the dominant form that is highly expressed in mammalian cells during embryogenesis or re-expressed in several

cancer types, including glioblastoma multiforme and lung cancer[79,80]. p42, the short form, is scarcely detectable in cancers, while its tumor-suppressive role has been established in several tumor models[80–82].

Similar to a *MYCN*-dependent neuroblastoma model[44], the genetic and chemical disruption of the PA2G4-MYC interface with the small molecule inhibitor WS6[47] altered AML cell proliferation, supporting an oncogenic role of PA2G4 in 3q26 AML. In AML, the mechanistic role of

**Fig. 4 | Effect of HDACis in human 3q26 EVI1^High^ AML. A** H&E histological sections of PR#003 bone marrow cells grown in collagen type I scaffolds with vehicle (DMSO) or HDACis (AR-42, belinostat, and entinostat) for 72 hr. Scale bar: 40 μm. **B** Percentage of cell viability of primary 3q26 AML grown in collagen type I scaffolds and treated with HDACis. Scale bar: 100 μm. Drug concentration (IC₅₀) used in (**A**) and (**B**) were established following a dose-response titration assay presented in Supplementary Fig. 6A. **C** AUC effect of HDACis and chemotherapy agents in EVI1^High^ (samples PR#002-008, and PR#023-024) and EVI1^Low^ (samples PR#010-022, and PR#025-039) AML blasts. **D** EVI1, ΔEVI1, and cleaved caspase 3 protein expression of primary 3q26 AML samples after 24 h of treatment with the indicated concentrations of HDACis. **E** Representative images of IF staining of primary 3q26 EVI1^High^ AML cell showing EVI1 nuclear content (in red) following 24 hr of treatment with DMSO or HDACis at the indicated concentrations (*n* = 3 biological replicates). Nuclei in blue (DAPI), scale bar: 100 μm. **F** Quantitative IF analysis of nuclear EVI1 content in 3q26 EVI1^High^ primary AML cells after 24 h of treatment with indicated compounds. **G** Effect of entinostat (left) or ara-C (right) on EVI1 nuclear localization (in red) following 6 hr of treatment in PR#003 or PR#004, respectively. The nuclei were stained with DAPI (blue). Scale bar: 100 μm. IF quantification of EVI1 nuclear content is presented at the bottom. **H** EVI1 expression (brownish) in bone marrow leukemia cells at diagnosis and following two cycles of azacitidine and entinostat (PR#002, left) or three cycles of ara-C and daunorubicin (3 + 7) (PR#004, right). Top, formalin-fixed, paraffin-embedded (FFPE) tissue sections were stained with anti-EVI1 antibody revealed by immunoperoxidase. Scale bar: 100 μm. Statistical significance was determined by a two-tailed non-parametric t-test (Mann-Whitney) (**C, G, H**) or one-way ANOVA with Tukey's correction for multiple comparison testing (**B, F**). Data are presented as mean ± SD in **B** (PR#003 *n* = 3, PR#004 *n* = 2), **C** (EVI1^High^ *n* = 9, EVI1^Low^ *n* = 28), **F** (PR#002 *n* = 318, PR#003 *n* = 573), **G** (PR#003 *n* = 235, PR#004 *n* = 214), and **H** (*n* = 10 fields per condition). Source data are provided as a Source Data file. See also Supplementary Fig. 5-6 and Supplementary Data 2.

PA2G4 has not yet been investigated. Previous studies demonstrated that PA2G4 is highly expressed in AML clinical samples compared to mononuclear cells from healthy donors[83]. However, since only a few cases have been cytogenetically characterized[83], it is impossible to speculate whether patients carrying 3q26 aberrancies express a higher level of PA2G4 than do other subgroups. Interestingly, *PA2G4* is part of a transcriptional core signature of four genes (*DNMT1*, *MYB*, *PA2G4*, and *YBX1*) repressed with de-differentiation in all-trans retinoic acid-induced NB4 and HL60 AML cell lines, suggesting that PA2G4 plays a role in AML maintenance[84]. Furthermore, in the murine myeloid precursor 32D cell line, PA2G4 participates in the regulation of HDAC8 ubiquitination that is disrupted by the forced expression of the *Cbfb-MYH11* fusion, indicating a contribution of PA2G4 in MDM2-mediated HDAC8 ubiquitination in inv(16) leukemia[59].

In conclusion, we propose a clinical treatment strategy that capitalizes on the *EVI1*-suppressing capacity of some HDACis in patients carrying 3q26 abnormalities that lack biomarker-directed treatment approaches. Furthermore, based on the mechanism driving efficacy upon HDAC inhibition, our work positions PA2G4 as a druggable target for a neglected population with EVI1-expressing AML and potentially for other Myc-dependent cancers.

## Methods
### Cell lines
The human cell lines MOLM1 (#ACC 720), UCSD/AML1 (#ACC 691), HNT34 (#ACC 600), TF1 (#ACC 334), MUTZ-3 (#ACC 295), 293 T (#ACC 635), OCI/AML3 (#ACC 582), MOLM13 (#ACC 554), NOMO1 (#ACC 542), OCI/AML2 (#ACC 99), GDM1 (#ACC 87), HL-60 (#ACC 3), SKM1 (#ACC 547) and 5637 (#ACC 35) were purchased from the Leibniz-Institut DSMZ-German collection of microorganisms and cell cultures (Germany). IMS-M2 were previously reported in[85]. U937T and U937T_E10 were a kind gift from the Rotraud Wieser laboratory (University of Vienna, Clinic of Medicine I, Waehringer Guertel 18-20, 1090 Vienna, Austria). Cells were cultured in RPMI 1640 (Thermo Fisher Scientific, Waltham, MA, USA, #31870074) with 10% or 20% fetal bovine serum (FBS) (Thermo Fisher Scientific, #10270106) and 1% penicillin-streptomycin (EuroClone, Pero-Milan, Italy, #ECB3001D). UCSD/AML1 medium was supplemented with 10 ng/mL rhGM-CSF (ProteinTech Group, Rosemont, IL, USA, #HZ-1002). MUTZ-3 cells were maintained in DMEM (Thermo Fisher Scientific, #11965118) with 20% FBS, 20% 5637 conditioned medium, and 2 mmol/L L-glutamine (Thermo Fisher Scientific, #25030-024). The tetracycline-repressible U937T and U937T_E10 were cultured in RPMI 1640 containing 10% tetracycline-free FBS (Takara Bio Europe SAS, Saint-Germain-en-Laye, France, #631367). 293 T cells were cultured in DMEM, 10% FBS, and 1% penicillin-streptomycin. Cell lines were grown in a humidified incubator at 37 °C and 5% CO₂, routinely identified by short tandem repeat profiling, and monitored for mycoplasma contamination.

### Clinical AML samples and patients
Leukemia cells derived from the peripheral blood and bone marrow of patients with 3q26 AML were obtained under approved protocols at the Department of Medicine and Surgery at Parma University Hospital (n. 18249/18/05/2017, and n. 29785/13/07/2021), according to the Declaration of Helsinki guidelines for the protection of human rights. Mononuclear cells were isolated by density gradient centrifugation using an LSM-lymphocyte separation medium (Cappel™ MP Biomedicals, Solon, OH, USA, #50494). We collected a total of 39 samples (Male = 18; Female=21; mean age = 64.4 -25-87-). Bone marrow biopsies were collected according to the protocol n. 265/2019, granted by the same Institution. All biological samples were collected after the release of a written informed consent. Three patients (PR#001, PR#002, and PR#003) were enrolled in a compassionate use program (the treatment schedule was described in the NCT00101179 clinical trial), they have provided written consent and they did not receive compensation. Individuals agreed to participate in our study and having their data reported for scientific purposes. No sex and/or gender was considered in the design of this study.

### Karyotype analysis and fluorescence in situ hybridization
Primary leukemic blasts were isolated as described in the previous section and cultured for 48 hr in RPMI 1640 in a humidified incubator at 37 °C and 5% CO₂. Cell media was supplemented with 0.1 μg/mL of colcemid (Thermo Fisher Scientific, #15212012) for 2 hr, followed by incubation in a hypotonic solution (0.075 M KCl). Cells were fixed in a 3:1 methanol (Sigma-Aldrich, St. Louis, MO, USA, #322415) and acetic acid glacial fixative solution (Sigma-Aldrich, #A6283) and spread on top of Superfrost Plus microscope slides (Thermo Fisher Scientific, #10149870). For the karyotype analysis, chromosome banding was performed by quinacrine (Q-banding) or Giemsa (G-banding) staining. A minimum of 20 metaphases per sample were acquired using a Nikon Eclipse 80i microscope (Nikon Instruments, Inc., Melville, NY, USA) and analyzed using NIS element software (Nikon Instruments, Inc.). For the fluorescence in situ hybridization (FISH) analysis, 10 μL of *MECOM/RUNX1* t(3;21) fusion (Kreatech Biotechnology B.V., Amsterdam, The Netherlands, # KBI-10310) or *MECOM* t(3;3); inv(3)(3q26) break-apart translocation probes (Kreatech Biotechnology B.V., # KBI-10204) were incubated at 37 °C for 12-16 hr after a phase of DNA dehydration with ethanol-scale incubation (75–85–100%) and DNA denaturation [75 ± 1 °C 5 minutes (min)]. Slides were washed once with 0.4x saline sodium-citrate/0.3% NP40 buffer at 73 ± 1 °C, followed by 4 x SSC/0,1% NP-40 at ambient temperature. DNA was counterstained with 4′,6-diamidino-2-phenylindole (DAPI, Sigma-Aldrich, #10236276001) before microscope analysis (Eclipse 80i microscope, Nikon Instruments, Inc.). Two-hundred interphase nuclei and at least 6 metaphases were analyzed for each patient, and DNA rearrangements were defined starting from a 5% cutoff for each probe.

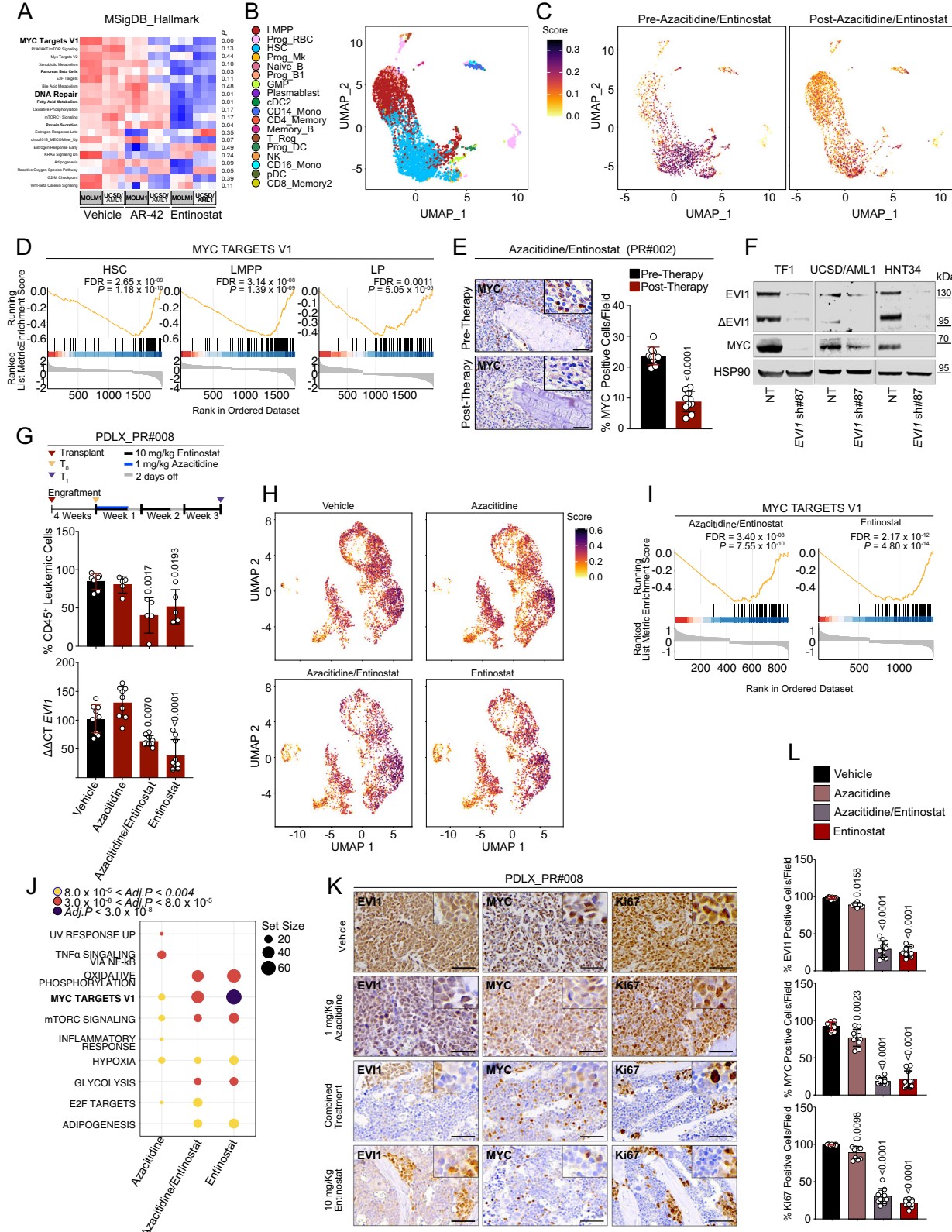

**Small molecule screening assay**

We screened 5349 compounds in duplicates, including i) the Spectrum Collection library (MicroSource Discovery System, Inc., Gaylordsville, CT, USA) containing 1200 FDA/EMA-approved drugs, 500 natural products, and 300 molecules in preclinical stages for a total of 2000 compounds; ii) the anti-cancer compound library (Selleck Chemicals, Houston, TX, USA), containing 349 bioactive compounds with known anti-cancer activity; and iii) the NDL-3000 library (TimTec, Tampa, FL, USA) containing 3000 natural derivative compounds. MOLM1 and UCSD/AML1 cells ($1.5 \times 10^3$ per well) were plated in 384-well tissue culture plates in 45 µL of medium using the BioTeck EL460 automated cell dispenser (BioTeck) and incubated for 72 h. Small molecules were screened at the final concentration of 1 µM dissolved in (dimethyl sulfoxide) DMSO added by a Tecan Evo200 (Tecan, Switzerland).

**Fig. 5 | HDAC-mediated suppression of EVI1 modulates Myc signaling.**
**A** Heatmap showing ssGSEA enrichment of MSigDB of gene signature in MOLM1 and UCSD/AML1 HDACi-treated cells. Hot or cold colors indicate correlation or anticorrelation of the top enriched gene sets from each functional group ($P \leq 0.05$ in bold, calcolated according to[30]) among cell lines and treatments (bottom).
**B** Uniform manifold approximation and projection (UMAP) plot of clustering results of PR#002 bone marrow cells before and after therapy. Colors indicate the cell populations on the basis of the reference mapping approach. **C** UMAP plot of clustering results of PR#002 bone marrow cells before (left) and after (right) therapy, colored according to the UCell score of leukemic markers. **D** GSEA running score plot of the top enriched *MYC targets V1 pathway* in HSC, LMPP, and LP ($Adj.P = 0.0011$) cell populations of PR#002. Each graph indicates the running enrichment score (ES) of the pathway (top), the location of single genes of the gene set in the ranking (central), and the distribution of the ranking metric (bottom).
**E** MYC expression (brownish) in bone marrow leukemia cells of PR#002 at diagnosis and following two cycles of azacitidine and entinostat (left). Scale bar: 100 μm. Right, the scatter dot blot indicates the mean ± SD of the percentage of the MYC-positive cells in FFPE tissue sections ($n = 10$ fields per condition). **F** EVI1, ΔEVI1, and MYC expression in EVI1^High AML cell lines, 6 days after shRNA transduction. NT = non-targeting, sh#87 = shRNA directed against *EVI1* (n = 3 biological replicates). **G** In vivo antileukemic effect of entinostat and azacitidine. Mice (PDLX_PR#008) were treated with vehicle (DMSO), azacitidine (1 mg/kg) for 5 days, entinostat (10 mg/kg) (5 days/week) or the combination of both for 3 weeks. Histograms show the percentage hCD45+ in bone marrow (top) and the percentage of *EVI1* mRNA relative to the control gene *RPL13A* (ΔΔCT) in CD45+ cells (bottom) at the end of treatment.
**H** UMAP plot of clustering results of PDLX_PR#008 bone marrow LP before (i) and

after entinostat (ii), azacytidine (iii) or the combination of both (iv), colored according to the UCell score of leukemic markers. **I** GSEA running score plot of the top enriched *MYC targets V1 pathway* in PDLX_PR#008 LP after the treatment with azacitidine/entinostat ($Adj.P = 3.40 \times 10^{-8}$) or entinostat as a single agent ($Adj.P = 2.17 \times 10^{-12}$). Each graph indicates the running enrichment score (ES) of the pathway (top), the location of single genes of the gene set in the ranking (central), and the distribution of the ranking metric (bottom). **J** Dot plot of GSEA results illustrating Molecular Signatures Database (MsigDB) biological processes associated with the indicated treatments compared to the vehicle in PDLX_PR#008 LP. Set size refers to the number of genes associated with each (MsigDB) biological process. Dot color indicates the range of *Adj.P* for each pathway. **K** EVI1, MYC, and Ki67 expression (brownish) in bone marrow leukemia cells (PDLX_PR#008) following three weeks treatment of azacitidine and entinostat as described in panel **G**. FFPE tissue were stained with anti-EVI1, anti-MYC and anti-Ki67 antibodies revealed by immunoperoxidase. Scale bar: 100 μm. **L** Histograms display the mean ± SD of the percentage of the EVI1, MYC or Ki67-positive cells in FFPE tissue sections. Statistical significance was determined by a two-tailed non-parametric t-test (Mann-Whitney) (**E**), and one-way ANOVA with Tukey's correction for multiple comparison testing (**G**, **L**). GSEA enrichment score significance was based on a weighted Kolmogorov Smirnov (WKS) test corrected for multiple hypotheses testing: Benjamini & Hochberg (BH or FDR) (**D**, **I**, **J**). Data are presented as mean ± SD in **E** ($n = 10$ fields per condition), **G** (vehicle $n = 7$, treated $n = 5$ per group; $n = 9$ for the bottom panel), **K** (vehicle $n = 7$, treated $n = 5$ per group), **L** ($n = 10$ fields per condition). Source data are provided as a Source Data file. See also Supplementary Fig. 7-8 and Supplementary Data 3.

---

DMSO and etoposide (5 μM) treated cells were used as negative (vehicle) and positive controls, respectively.

The effect on cellular viability was quantified using an ATP-based Cell Titer-Glo assay (Promega Corporation, Madison, WI, USA, #G7573), and presented as a percentage of viable cells over control (POC). We also defined the normalized percentage of cell death (ΔPOC) on the basis of the average of duplicates (POC_molecules, Supplementary Data 1) using the following formula: [negative controls luminescence (DMSO)−sample luminescence]/[negative controls luminescence (DMSO)] x 100. POC_molecule values for each compound have been compared to the DMSO control values ($n = 24$) using an ANOVA test applied to a linear regression model. The resulting p-values were adjusted for the multiple comparisons using the Benjamini & Hochberg (BH or FDR) method included in the p.adjust function implemented in the stats R package. We prioritized hits on the basis of the 95th percentile limit of the highest statistically significant ΔPOC [Benjamini-Hochberg $P \leq 0.05$]. We then calculated all pairwise distances between the given fingerprints of the top candidates and fit a beta distribution to the resulting Tanimoto scores, conditioned on the number of set bits in each fingerprint (ChemmineR package). The resulting matrix was clustered using the hclust function in the stats package with Euclidean distance and the average agglomeration method.

Forty compounds listed in Supplementary Data 2 were counter-screened in additional cell lines (HNT34, NOMO1, and THP1) in a six-log [concentrations] dilution ranging from 10 μM to 30 nM. The tertiary screen in TF1 assayed 4942 small molecules from the annotated libraries derived from the Selleck and Sigma-Aldrich collection available at the screening unit of the Leibniz-Forschungsinstitut of Molecular Pharmacology (FMP) and listed in Supplementary Data 1. BRAID analysis of combined drug action was calculated according to[86].

## High-throughput in silico screening
Marker genes for the *EVI1 "On"* vs. *"Off"* signature were chosen using publicly available Affymetrix microarray expression profiling data on TF1 cells transduced with shRNAs targeting *EVI1* (E-GEOD 16238) or from mRNA quantification in AML cell lines harboring 3q26 aberration (E-MTAB-2225)[15]. For the TF1 study, we then inferred marker genes to the ConnectivityMap[20,87] (CMap) database (https://clue.io/lincs), while

for the cross-validation set in *EVI1* repressed HNT34 or AML^High vs. AML^Low with the The Library of Integrated Network-Based Cellular Signatures (LINCS)[88]. We used the parametric bootstrap method on sets of molecules to calculate the enrichment of compound classes[89]. CMap and LINCS/L1000FWD are accessible at https://clue.io/lincs and https://lincsproject.org/, respectively.

## Cell treatment and viability assay
AR-42 (#S2244), belinostat (PXD101, #S1085), entinostat (MS-275, #S1053), cytarabine (ara-C, #S1648), vincristine sulfate (#S1241), daunorubicin HCl (#S3035), methotrexate (#S1220), WS6 (#S7442), ( + )-JQ-1 (#S7110) and MG132 (#S2619) were obtained from Selleck Chemicals. Cell solution (50 μL/well of $0.02 \times 10^6$/mL) was dispensed in 384-well plates (Corning Life Sciences Plastic, Bedford, MA, USA, #3570) using Multidrop™ Combi (Thermo Fisher Scientific, #5840300). According to the manufacturer's protocol, small molecules were dissolved in DMSO or ddH$_2$O and added with a nanometric Tecan D300e dispenser (Tecan Trading AG, Switzerland) at the concentration indicated in the figures. Cell viability was assessed after 72 hr of drug treatment using a CellTiter-Glo ATP assay (Promega Corporation, #G7573). Values for IC$_{50}$ and the area under the curve (AUC) were calculated using GraphPad Prism 8 software (La Jolla, CA, USA).

## Apoptosis and DNA content assays
Apoptosis was measured by annexin V (annexin V, FITC conjugate, Thermo Fisher Scientific, #A13199) and propidium iodide (Thermo Fisher Scientific, #BMS500PI) staining. Cells were analyzed by flow cytometry with a FACScan (Beckman Coulter-Cytomics FC 500, Brea, CA, USA) or Attune NxT (Thermo Fisher Scientific) flow cytometry, and data were processed by FlowJo V10 (Tree Star, LLC, Ashland, OR, USA) analytical software. Cellular DNA content was assessed by staining with propidium iodide (50 μg/mL). At least 20,000 events were acquired, and all of the determinations were replicated at least twice.

## Methylcellulose assay
Clonogenic assays were performed using the MethoCult™ colony-formation assay (STEMCELL Technologies, Vancouver, BC, Canada, #GFH84444). In brief, a 10x concentrated AML cell suspension was

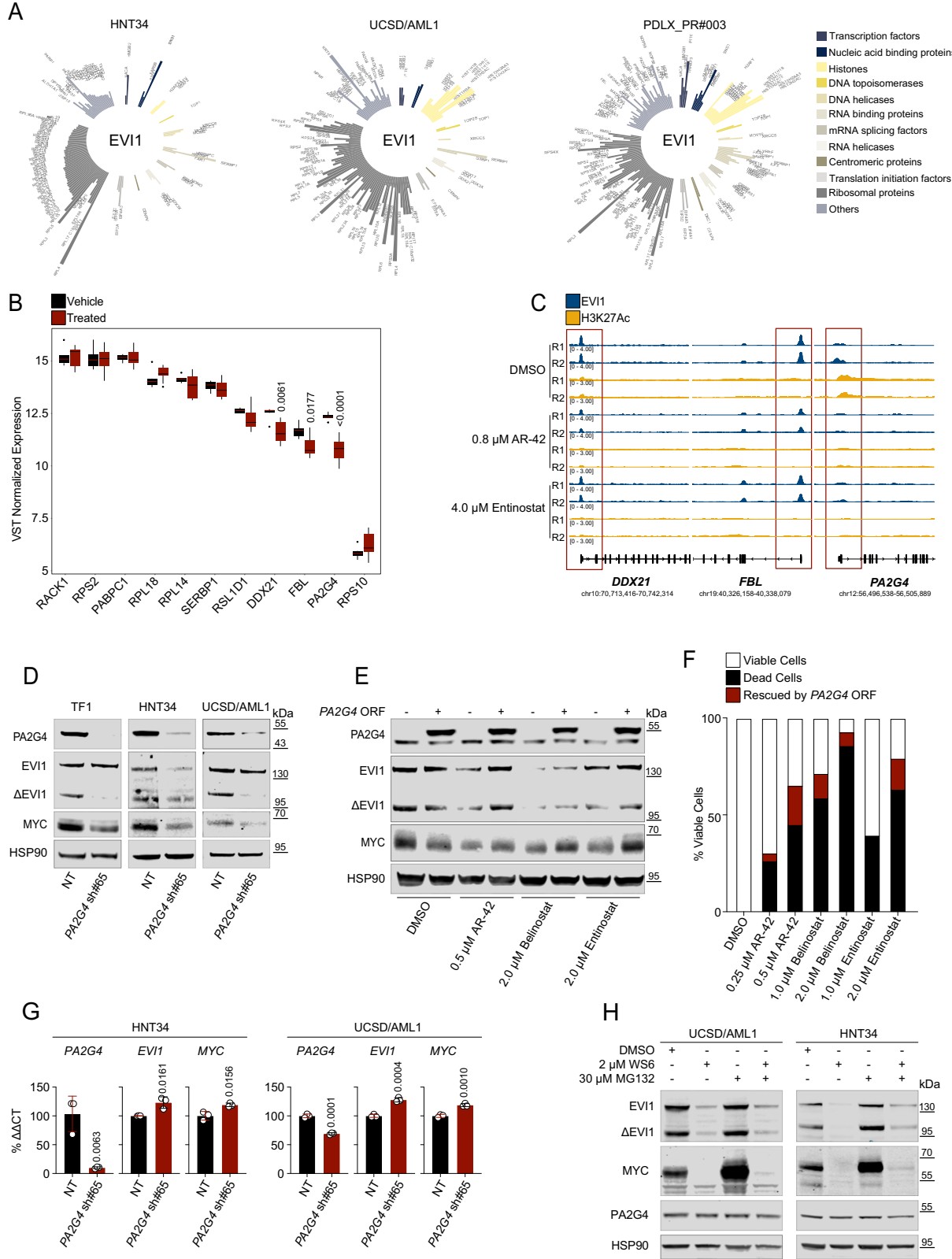

prepared and mixed with thawed MethoCult™ in a 1:10 v/v ratio of cells: MethoCult™. Samples were vortexed thoroughly and left to stand for at least 5 min to allow bubbles to rise to the top before dispensing; they were then incubated at 37 °C, in 5% $CO_2$, with 95% humidity for 15-20 days. Colony identification and counting were performed using an EVOS FL inverted microscope (Thermo Fisher Scientific) equipped with 4x and 10x objectives.

**Immunodetection and antibodies**

Whole protein lysate was extracted using 1x cell lysis buffer (Cell Signaling Technology, Danvers, MA, USA, #9803 S) supplemented with phospho-stop phosphatase inhibitor (Sigma-Aldrich, #04906837001) or protease/phosphatase inhibitor cocktail 100x (Cell Signaling Technology, Danvers MA, USA, #58725) and Complete Mini, EDTA-free protease inhibitor (Sigma-Aldrich, #11836170001). Cells were lysed on

**Fig. 6 | PA2G4 mediates the effect of HDACis on EVI1 and MYC. A** Circle plot showing chromatin-associated proteins immunoprecipitated with an anti-EVI1 antibody by RIME. The bar length indicates the mean of mass spectrometry spectral count (SPEC) of uniquely identified proteins of two biological replicates per condition. Bar colors indicate protein ontology of the EVI1 interactors. **B** Histograms show the normalized gene expression levels of *SERBP1, RPL14, RPL18, RPS2, PA2G4, RPS10, FBL, PABPC1, RACK1, RSL1D1,* and *DDX21* following 16 hr of 0.5 μM AR-42, 2 μM entinostat in UCSD/AML1, or 0.8 μM AR-42 and 4 μM entinostat in MOLM1 compared to vehicle. Statistical significance was determined using DESeq2. Whiskers show median values (central black lines) and 25th and 75th percentiles (bottom and top bounds), respectively. The bars represent values that exceed 1.5 times the interquartile range (IQR) from the edge of each box: (vehicle *n* = 6, treated *n* = 12). **C** Tracks showing EVI1 binding and H3K27ac enrichment across the *DDX21, FBL, and PA2G4* locus in HNT34 cells treated with vehicle (DMSO), AR-42 and entinostat. The bottom bar represents the genes (hg19), and the y-axis represents normalized read density scaled to 1 million. **D** PA2G4, EVI1, ΔEVI1, and MYC expression in TF1, HNT34, and UCSD/AML1 cells 6 days after shRNA transduction (*n* = 3 biological

replicates). **E** Effect of *PA2G4* overexpression on HNT34 cells treated with HDACis. Western blot analysis showing PA2G4, EVI1, ΔEVI1, and MYC in wild-type or *PA2G4*-overexpressing HNT34 cells after 24 h of treatment with HDACi at the indicated doses (*n* = 2 biological replicates). (**F**) Histogram indicates the percentage of viable (white), dead (black), or rescued (red) HNT34 cells (*n* = 20.000) by the over-expression of *PA2G4* on the basis of positivity for annexin V/PI staining after 72 h of HDACis treatment (*n* = 2 biological replicates). **G** Percentage of *PA2G4, EVI1,* and *MYC* mRNA relative to the control gene *ACTB* (ΔΔCT) in HNT34 and UCSD/AML1 cells 6 days after shRNA transduction. Data are presented as mean ± SD (*n* = 3). Statistical significance was determined by a two-tailed unpaired, parametric t-test. **H** Proteasome inhibition rescues WS6-induced EVI1 and MYC protein degradation. EVI1, ΔEVI1, MYC expression, and PA2G4 in HNT34 and UCSD/AML1 cell lines after treatment with DMSO and WS6 (24 hr) or MG132 (4 hr) at the indicated concentrations (*n* = 2 biological replicates). NT = non-targeting, sh#65 = shRNA directed against *PA2G4* in **C** and **F**. Source data are provided as a Source Data file. See also Supplementary Fig. 9-10 and Supplementary Data 4.

ice for 10 min with gentle stirring and centrifuged at 21,130x *g* for 10 min at 4 °C. Protein assay dye reagent (Biorad Laboratories, Hercules, CA, USA, #5000006) was used for protein quantification, and 40 μg of total lysate/sample was loaded for SDS-PAGE analysis. The following specific antibodies were purchased from Cell Signaling Technology: anti-EVI1 (C50E12, #2593), β-actin (8H10D10, #3700), cleaved caspase 3 (Asp175, #9661), and MYC (#9402). Anti-PA2G4 antibodies were purchased from Sigma-Aldrich (#HPA016484, #SAB1402863), Abcam (Cambridge, United Kingdom, #ab180602) and Proteintech (Rosemont, IL, USA #66055-1-Ig, #15348-1-Ap). Anti-histone H3 (acetyl K27) was purchased from Abcam (#ab4729). Anti-acetylated α-Tubulin (#sc-23950) and HSP90 (#sc-69703) were purchased from Santa Cruz Biotechnology (Dallas, TX, USA). All primary antibodies were used at a dilution factor of 1:1000. IRDye 680LT goat anti-mouse IgG (#925-68020), IRDye 800CW goat anti-rabbit IgG (#925-32211), and IRDye 680RD goat anti-rabbit IgG (#925-68071) from LI-COR Biotechnology (Lincoln, NE, USA) were used as secondary species–specific antibodies at a dilution factor of 1:10000. Signals were detected using the LI-COR Odyssey imaging system (LI-COR Biotechnology).

### RNA processing and quantification

RNA was extracted using the RNeasy mini kit (Qiagen, Hilden, Germany, #74106). According to the manufacturer's protocol, cDNA was synthesized using the high-capacity cDNA reverse transcription kit (Thermo Fisher Scientific, #4368814), using 1 μg of RNA as starting material. A quantitative real-time polymerase chain reaction (qPCR) was performed using TaqMan gene expression assays for *MECOM* (Thermo Fisher Scientific, #Hs00602795_m1), *MYC* (Thermo Fisher Scientific, #Hs00153408_m), *PA2G4* (Thermo Fisher Scientific, #Hs00854538_g1), and TaqMan™ Universal PCR master mix (Thermo Fisher Scientific, #4364338) in an Applied Biosystems™ StepOne™ real-time PCR system (Thermo Fisher Scientific, #4376357). Each condition was run in triplicate. The expression levels of the target genes were normalized to those of *RPL13A* or *ACTB* (Thermo Fisher Scientific, #Hs04194366_g1, #Hs99999903_m1). Data were analyzed using the ΔΔ cycle threshold (CT) method and plotted as a percentage relative to control.

### Immunofluorescence

Cells (50 × 10³) were washed in PBS and spotted on Superfrost Plus microscope slides (Thermo Fisher Scientific, #10149870) using a cytospin centrifuge (CR2000 Small Prime Centrifuge, Centurion). Fixation was carried out for 10 min in PBS, 4% paraformaldehyde (Thermo Fisher Scientific, #28908) at 4 °C. For nuclear protein staining, cells were permeabilized for 10 min in PBS and 0.4% Triton X-100 (Sigma-Aldrich, #T-9284) at ambient temperature. We applied a

blocking solution for 1 h at ambient temperature composed of 5% bovine serum albumin, 0.1% Triton X-100, and 1% goat serum (Abcam, #ab138478) diluted in PBS. Cells were then incubated for 1 hr at ambient temperature with the antibody targeting EVI1 (Cell Signaling Technology, #2593, 1:1000 dilution) or PA2G4 (Sigma-Aldrich, #HPA016484, #SAB1402863 both diluted 1:1000) and subsequently revealed by secondary antibodies (Invitrogen, Carlsbad, CA, USA, #A11029, #A11036, both diluted 1:400) and diluted in blocking solution. Nuclei were stained with DAPI (Sigma-Aldrich, #D9542). Prolong Gold Antifade reagent (Thermo Fisher Scientific, #P36934) was used as a mounting solution. Images were captured using an EVOS™ M5000 microscope (Thermo Fisher Scientific) or a Leica Stellaris 5 confocal microscope. ImageJ (http://rsbweb.nih.gov/ij/) and Leica Image Compass were used for the analysis.

### Immunohistochemistry

Informed consent was obtained from patients undergoing trephine bone marrow biopsy from the iliac crest for diagnostic procedures according to the ethical guidelines and the protocol 265/2019/TESS/UNIPR, approved by the University of Parma. Following decalcification, 4-μm-thick sections obtained from formalin-fixed, paraffin-embedded bone marrow biopsies were processed for immunohistochemical staining. Bone marrow samples were deparaffinized and rehydrated in decreasing alcohol scale; enzymatic epitope retrieval was performed by water bath at 95 °C, 40 min, in ULTRA Cell Conditioning Solution (Ventana; 950-224); endogenous peroxidase was blocked with 3% $H_2O_2$. After incubation with rabbit anti–EVI1 (Cell Signaling Technology, #2593; 1:400 dilution; overnight; 4 °C), rabbit anti-MYC (Cell Marque™; Rocklin, CA, USA, #395R-18; ready to use; 40 min; 37 °C), or mouse anti-Ki67 (Agilent, Santa Clara, CA, USA, #IR626; ready to use; 30 min; 37 °C) antibodies, all sections were revealed by IHC Detection Kit-Micropolymer (Abcam #ab236466) according to manufacturer's recommendations. The nuclei were counterstained by light hematoxylin. All staining steps were performed at ambient temperature. 3D type I collagen scaffolds and xenografted tumors were fixed in formalin, included in paraffin, and processed as indicated above. Images were acquired with a MoticEasyScan One (Motic, Hong Kong) microscope and quantified using QuPath software v.0.3.2 (https://qupath.github.io/)[90].

### Chromatin immunoprecipitation and ChIP Sequencing

Chromatin was collected from HNT34 cells (20 × 10⁶ cells each condition) using ActiveMotif ChIP-IT® Express (ActiveMotif, Carlsbad, CA, USA, #53008) according to the manufacturer's instructions and sheared in 12 sets of 10-second sonication using a Branson Digital Sonifier (Thermo Fisher Scientific). Chromatin was incubated overnight at 4 °C with 10 μg of the following antibodies: EVI1 (Cell Signaling Technology, #2593 S), MYC (Cell Signaling Technology,

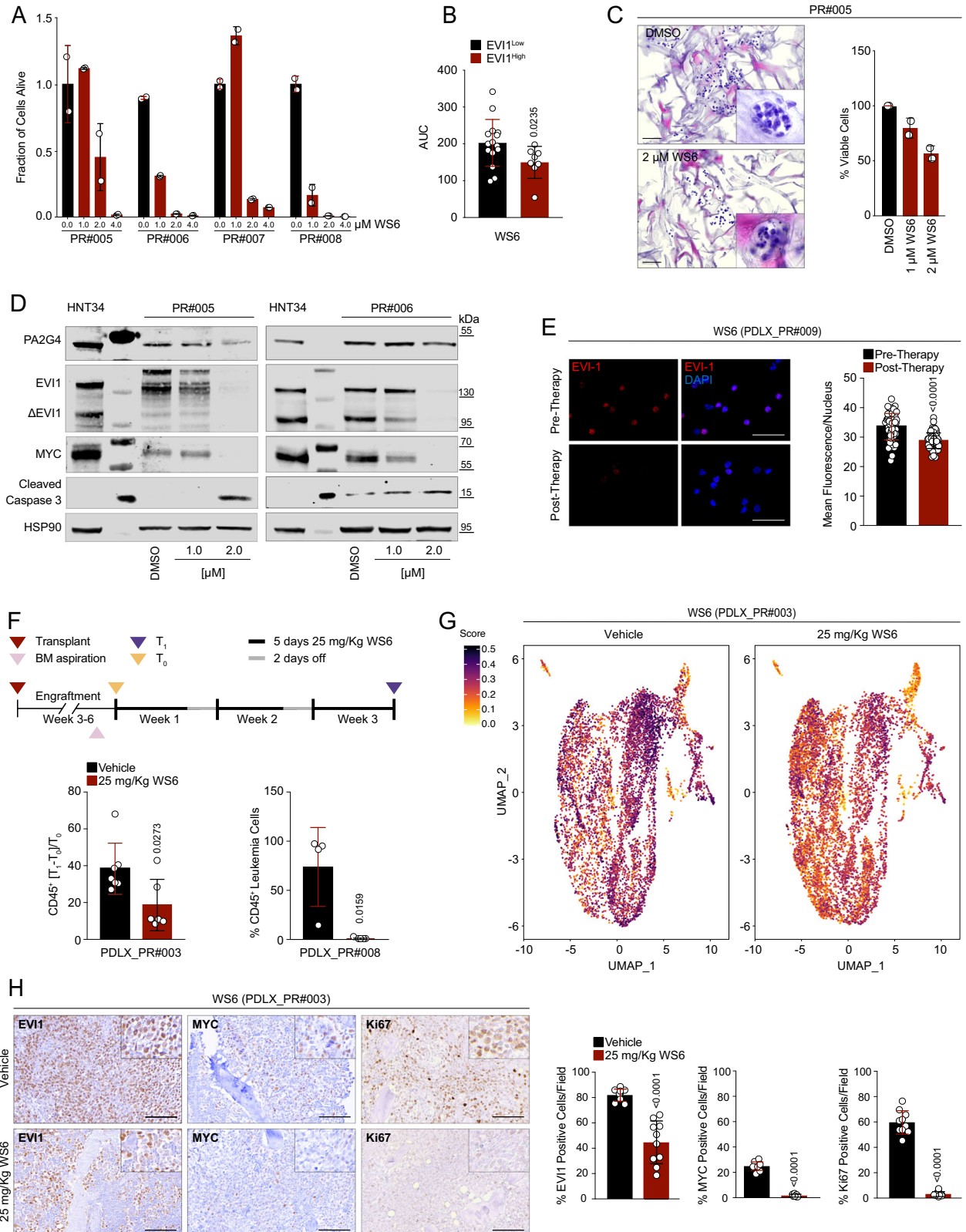

#9402 S), histone 3 at lysine 27 (H3K27Ac, Abcam, #ab4729), and rabbit IgG (Merck, #12-370). DNA was purified with AMPureXP beads (Beckman Coulter, #A63881) following the manufacturer's protocol. Real-time PCR was performed with SYBR-Green Reagents (Thermo Fisher, #4309155) on a CFX96 Real-Time PCR System (Biorad) with primers targeting the E-Box DNA binding sites in the *PA2G4* gene promoter, 500 bp upstream of the transcription starting site

region (forward: CCTCCCCGACCTAGGTGTA; reverse: GCTGAGC-GAGAGCCAGTAAC) and *MYC* gene promoter (forward: AGGGT-GAGGTCAAGCATTTG; reverse: TGGCCTTGAACCCATACTTC). Fold enrichment was calculated by dividing the input-normalized CT by the negative IgG control CT.

For Chromatin Immunoprecipitation Sequencing (ChIP-Seq), purified DNA was processed using the NEBNext® Ultra™ II DNA Library

**Fig. 7 | PA2G4 suppression alters 3q26 AML proliferation in vivo. A** Effects of WS6 treatment on viability in primary 3q26 AML samples following 72 hr of drug treatment. **B** Effect of WS6 in EVI1$^{High}$ (samples PR#002 - 006, PR#008 and PR#023 - 024) and EVI1$^{Low}$ (samples PR#025-039) primary AML samples. The AUC model of the log-transformed dose-response data is depicted. **C** Effects of WS6 treatment compared to control (DMSO) on viability in primary 3q26 AML samples following 72 hr of drug treatment. Left: H&E of PR#005 bone marrow cells in 3D cell culture after 72 hr of treatment with vehicle (DMSO) or WS6 at 1/2 IC$_{50}$ and IC$_{50}$ concentrations. Scale bar: 40 μm. Right: histograms indicate the fraction of viable cells expressed as a percentage relative to control. **D** PA2G4, EVI1, ΔEVI1, and MYC and cleaved caspase 3 protein expression in primary 3q26 AML samples following 24 h of treatment with the vehicle (DMSO) or WS6 at the indicated concentrations. **E** Effect of WS6 (50 mg/kg) on EVI1 nuclear localization (in red) following 6 hr of treatment in PDLX_PR#009 (n = 3 biological replicates). Nuclei in blue (DAPI). Scale bar: 100 μm. Histograms indicate the fluorescence intensity of EVI1 nuclear content before and after treatment. **F** In vivo antileukemic effect of WS6. On the top the draft of the experiments. Mice were treated with vehicle (DMSO) or 25 mg/Kg WS6 for 5 days/week for 15 days. On the bottom-left dot plot showing the number of hCD45+ cells expressed as the percentage difference of BM leukemic cells at T1

(endpoint-day 15) vs. T0 (pre-treatment-day 0) normalized for T0 in control (n = 7) and WS6-treated PDLX_PR#003 (n = 6). On the bottom-right the dot plot shows the percentage of hCD45+ cells in the bone marrow between vehicle (n = 4) or WS6-treated PDLX_PR#008 (n = 5) at the endpoint. Events ≥ 20.000. **G** UMAP plot of clustering results of PDLX_PR#003 hCD45 bone marrow–positive cells before (left panel) and after (right panel) 15 days of WS6 treatment (25 mg/kg/IP/5 days a week), color-coded according to the UCell score of leukemic markers (MECOM, MYC, CD45, CD34, KIT, CD33, ANPEP, CD38, CD2, TFRC, HLA-DRA, HLA-DRB1, and HLA-DRB5). **H** EVI1, MYC, and Ki67 expression in PDLX_PR#003 after 15 days of WS6 treatment (25 mg/kg/IP/5 days a week) or vehicle (DMSO). Scale bar: 100 μm. Histograms to the right indicate the mean ± SD of the percentage of the EVI1, MYC, or Ki67-positive cells in FFPE tissue sections. Statistical significance was determined by a two-tailed non-parametric t-test (Mann-Whitney) (**B, E, F, H**). Data are presented as mean ± SD in **A** (n = 2), B (EVI1$^{High}$ n = 8, EVI1$^{Low}$ n = 15), **C** (n = 2), in **E** (n = 125), **F** (vehicle PDLX_PR#003 n = 7, treated PDLX_PR#003 n = 6, vehicle PDLX_PR#008 n = 4, treated PDLX_PR#008 n = 5) and **H** (vehicle n = 7, treated n = 6; n = 10 fields per condition). Source data are provided as a Source Data file. See also Supplementary Fig. 11.

Prep Kit (#E7645, New England Biolabs, Ipswich, MA). The final libraries were paired-end sequenced using a 150-cycle kit on NovaSeq 600 (Illumina) with a sequencing depth of 50 million reads per samples. Sequencing data were analyzed using the nf-core ChIP pipeline (version 2.0.0)[91] that utilizes MACS2 for peak calling[92] and HOMER for annotation of peaks[93]. Normalized BigWig files were scaled to 1 million mapped reads to be able to compare coverage across multiple samples. Tracks illustrating read coverage and representative peaks were visualized using the IGV genome browser with Human (GRCh37/hg19) genome[94]. A range of ± 2 kb of the transcriptional start site (TSS) was retained to define promoter-specific peaks.

## Co-Immunoprecipitation

For the endogenous co-immunoprecipitation, HNT34 cells were lysed in modified Cell Lysis Buffer II (Invitrogen #FNN0021) supplemented with Protease and Phosphatase Inhibitors (Sigma #11836153001 and #04-906-837-001) for 30 min on ice and centrifuged at 14,000 RPM for 10 min at 4 °C. The protein lysates were then incubated overnight with the following antibodies: EVI1 (Cell Signaling Technology, C50E12 #2593), PA2G4 (Proteintech 15348-1-AP), or Normal Rabbit IgG (Merck, #12-370), at a concentration of 1 μg/mg of IP reaction. Dynabeads protein G magnetic beads (Thermo Fisher, #10003D) were added for 2 hr at 4 °C the next day, and then the purified proteins were eluted in 40 μl of 1X L.B. For the overexpression in 293 T cells, cells were transfected with Lenti-hEFI1a-ORF-P2A-eGFP-IRES-Puro expressing *MECOM* or *PA2G4* ORF. 48 hr after transfection, the cells were processed similarly Co-IP experiment described above.

## Scaffold-based 3D AML culture

To synthesize collagen scaffolds, an acid suspension of type I collagen was precipitated at pH 5.5 and cross-linked with 1,4-butanediol diglycidyl ether. The scaffold's porosity was generated by a freezing and heating ramp to ensure desired pore size, interconnectivity, and orientation. The materials were sterilized in 70% ethanol for 1 hr and washed in sterile Dulbecco phosphate-buffered saline (Life Technologies, Carlsbad, CA, USA). Scanning electron microscopy was used to characterize the scaffold macrostructure and microstructure. In brief, scaffolds were fixed in 2.5% glutaraldehyde 0.1 M sodium cacodylate buffer for 2 hr at 4 °C, dehydrated in a series of ethanol, dried in a desiccator overnight, and sputter-coated with platinum. Images were acquired with a Nova NanoSEM 230 (FEI, Hillsboro, OR, USA). The human AML cell lines HNT34 and MOLM1 were seeded in 2 × 9-mm scaffolds at concentrations of 0.25 × 10⁶, 0.5 × 10⁶, and 1 × 10⁶ cells. Seeding was reached by soaking 50 μL of cell suspension in dry scaffolds placed in a multi-well plate. Cells were allowed to adhere for 1 hr

at 37 °C before adding the culture medium. After 24 hr, the scaffolds were gently moved onto a new multi-well plate. The medium was replaced every 2-3 days. Cell growth was monitored after 1, 3, and 7 days by MTT assay. Scaffolds were then incubated with 1 mg/mL of MTT solution (Sigma-Aldrich) in culture medium for 2 hr at 37 °C. Cell viability was determined by reading the absorbance at 550 nm. The scaffolds were fixed in neutral buffered formalin, dehydrated in scaling ethanol solutions, and embedded in paraffin to assess cell morphology and distribution. Hematoxylin-and-eosin staining was performed in 5-μm-thick scaffold sections mounted onto Superfrost Plus microslides (Thermo Fisher Scientific, #10149870). Drug treatment was performed after 72 hr from seeding. Drug efficacy was evaluated by MTT assay, as described above. Results were presented as a percentage of cells alive relative to vehicle-treated cells. The drug concentrations that reduced cell viability to 50% of vehicle controls (IC$_{50}$) were calculated using GraphPad Prism 8 software (La Jolla, CA, USA).

## Virus production and transduction of AML cell lines

For virus production, 3 × 10⁶ 293 T were transfected with 2 μg of pCMV-VSV-G and pCMV-deltaR8.91 vectors (kind gift from Kymberly Stegmaier laboratory, Dana Farber Cancer Institute, Boston, MA, USA). pZIP-hEF1a-RFP-Puro vector (Transomic, Huntsville, AL, USA, #TLHSU1444) harboring the shRNA sequences listed in Table 1 was used for genetic downregulation experiments.

The Lenti-hEF1a-ORF-P2A-eGFP-IRES-Puro (Transomic, Huntsville, AL, USA, #TLO2015.1) vectors containing the *MECOM* (NM_005241.1), *PA2G4* (Gene BankBC001951.1) and *MYC* (NM_002467.6) open reading frame (ORF) sequences were used for genetic overexpression experiments. Cells were transfected using the FUGENE6 protocol (Promega Corporation, #E2691), and viral supernatant was harvested and filtered (0.4 μm) after 72 hr. AML cells were resuspended at a concentration of 4 × 10⁶ cells per 1 mL of serum-free RPMI and spin-infected for 1 h at ambient temperature with 100 μL of lentivirus particles and 8 μg/mL polybrene (Sigma-Aldrich). Puromycin (1 μg/mL) was added for selection (Thermo Fisher Scientific, #A1113803).

## Rapid immunoprecipitation mass spectrometry of endogenous proteins (RIME)

For chromatin immunoprecipitation, cells were fixed with 1% methanol-free formaldehyde (Cell Signaling Technology, #12606) for 8 min and quenched with 0.125 M glycine (Sigma-Aldrich, #G8898). Chromatin was isolated by adding lysis buffer, followed by disruption with a Dounce homogenizer (ActiveMotif, #40401). The lysates were sonicated, and the DNA sheared to an average length of 300-500 bp. Genomic DNA (input) was prepared by treating aliquots of chromatin

**Table 1 | Short hairpin RNA sequences for genetic knockdown of *EVI1* (shRNA#16, shRNA#87and shRNA#88), *PA2G4* (shRNA#65, shRNA#66 and shRNA#68) and Not Targeting control**

| Gene | ID | Sequence |
|---|---|---|
| *EVI1* | shRNA#16 | 3'TAAATTTCTCTTTATCACTTTC 5'AAAAGTGATAAAGAGAAATTTA Loop:TAGTGAAGCCACAGATG |
| *EVI1* | shRNA#87 | 3'TATATCATTGTCTTCATCCTCC 5'AGAGGATGAAGACAATGATATA Loop:TAGTGAAGCCACAGATGTA |
| *EVI1* | shRNA#88 | 3'TTCTTCAACTTCTTCATCATCC 5'AGATGATGAAGAAGTTGAAGAA Loop:TAGTGAAGCCACAGATGTA |
| *PA2G4* | shRNA#65 | 3'TAGTGGTTCTCTGTCCTGCATC 5'AATGCAGGACAGAGAACCACTA Loop:TAGTGAAGCCACAGATGTA |
| *PA2G4* | shRNA#66 | 3'TTCATGGTCCTTCTTCTGCTGG 5'ACAGCAGAAGAAGGACCATGAA Loop:TAGTGAAGCCACAGATGTA |
| *PA2G4* | shRNA#68 | 3'TAATTCAGCTTTTTCATGGTCC 5'AGACCATGAAAAAGCTGAATTA Loop:TAGTGAAGCCACAGATGTA |
| Not Targeting | | 3'ATGCTTTGCATACTTCTGCCTG 5'AAGGCAGAAGTATGCAAAGCAT Loop:TAGTGAAGCCACAGATGTA |

with RNase, proteinase K, and heat for de-crosslinking, followed by ethanol precipitation. Pellets were resuspended, and the resulting DNA was quantified on an ND-2000 NanoDrop spectrophotometer (Thermo Fisher Scientific). Extrapolation to the original chromatin volume allowed quantitation of the total chromatin yield. An aliquot of chromatin (150 μg) was pre-cleared with protein G agarose beads (Thermo Fisher Scientific, #15920010). Proteins of interest were immunoprecipitated using 15 μg of antibody targeting EVI1 (Cell Signaling Technologies, cat. #2593 S) and protein G magnetic beads (Thermo Fisher Scientific, #88847). Protein complexes were trypsinized to release the immunoprecipitate from the beads and digest the protein samples. Protein digests were purified using a C18 spin column (Harvard Apparatus, Holliston, MA, USA, #74-7242). The peptides were vacuum-dried using a SPD130XL SpeedVac (Thermo Fisher Scientific).

## Mass spectrometry

Digested peptides were analyzed by LC-MS/MS on a Q Exactive Orbitrap mass spectrometer linked to Dionex Ultimate 3000 HPLC and a nanospray Flex™ ion source (Thermo Fisher Scientific). Samples were loaded directly onto the separation column BEH C18, 75 μm x 100 mm, 130 Å 1.7-μm particle size (Waters Corporation, Milford, MA, USA). Peptides were eluted using a 100-min gradient with a 323 nL/min flow rate. An MS survey scan was obtained for the m/z range 340-1600, and MS/MS spectra were acquired using a top 15 method, where the top 15 ions in the MS spectra were subjected to high-energy collisional dissociation. An isolation mass window of 1.6 m/z was used for the precursor ion selection, and a normalized collision energy of 27% for fragmentation. A 20-second duration was used for the dynamic exclusion.

Tandem mass spectra were extracted and analyzed by PEAKS Studio version 8 build 20. Charge state deconvolution and deisotoping were not performed. The database consisted of the Uniprot database (version 180508, 71,771 curated entries) and the cRAP database of common laboratory contaminants (www.thegpm.org/crap; 114 entries). The database was searched with a fragment ion mass tolerance of 0.02 Da and a parent ion tolerance of 10 parts per million. Post-translational variable modifications consisted of methionine oxidation, asparagine, and glutamine deamidation. Peaks studio built-in decoy sequencing and FDR determination (decoy fused method) with a cutoff set to $-10\log P > 20$ was used to validate MS/MS-based peptide and the parsimony rules for protein identifications. A threshold of $\geq -10\log P$ of 20 was applied for peptide identifications. The weighted sum of nine parameters for peptide scoring is converted to a $P$ value, representing the probability of false identification. Protein identifications were accepted if they could pass the $-10\log P$ of 20 and contained at least one identified unique peptide. Proteins that contained similar peptides and could not be differentiated on MS/MS analysis alone were grouped to satisfy the principles of parsimony. Proteins sharing significant peptide evidence were grouped into protein groups.

The hit list was generated by taking all proteins (sample vs. IgG control) with a spectral count of five and above from each replicate.

For a label-free quantitation (LFQ) analysis, we included technical and biological replicates ($n = 6$) in a group named "3q26 AML" and compared them to the group including all the correlative IgG controls ($n = 6$), named "Control". The Thermo raw files were analyzed using MaxQuant (MQ) version 2.4.2.0. The LFQ intensities of proteins from MQ analysis were imported and filtered for reverse identifications (false positives), contaminants, and proteins "only identified by site". Data were transformed to $\log_2$ scale. Then, we imputed missing values and replaced them from a normal distribution. The protein quantification and calculation of statistical significance were performed with a two-sample t-test and with a permutation-based correction controlled with an FDR threshold of 0.05. A protein was considered as EVI1's interactor if the difference between the "3q26 AML" and "Control" groups was statistically significant ($P < 0.05$), the fold change was 4, and it was identified with a minimum of two peptides.

## RNA sequencing

PolyA-enriched, strand-specific RNA libraries were generated with the TruSeq mRNA stranded sample preparation kit (Illumina, San Diego, CA, USA, #20020594) starting from 1 μg of RNA from each sample. In brief, RNA was subjected to polyA selection using magnetic oligo-dT beads (Thermo Fisher Scientific, #61002). A 94 °C-incubation for 4 min partially fragmented polyA RNA. Both actinomycin-D (during the first-strand cDNA synthesis) and dUTP (during the second-strand synthesis) were used to keep the strand information. The libraries were end-repaired and adenylated before being ligated with Y-shape single-indexed adapters and amplified by 11 PCR cycles. The last purification step was performed with Ampure XP beads (Beckman Coulter, #A63882) at 0.8x to remove all adapter dimers. Each library was quantified and quality-controlled using a Qubit 4 fluorometer (Thermo Fisher Scientific) and LabChip GX (Perkin Elmer, Waltham, MA, USA). The adaptor-tagged pool of libraries was loaded onto Illumina Hiseq2500 rapid-run flow cells (SR100 chemistry) for cluster generation and deep sequencing. The raw sequence files were quality-controlled using FastQC (v 1.3) (http://www.bioinformatics.babraham.ac.uk/projects/fastqc/, accessed in May 2020). Transcripts were aligned using the STAR package (v2.7.1a) and quantified using the "quantMode GeneCounts" function with ENSEMBL annotation and the human genome version GRCh38 as a reference. Read counts generated by STAR were analyzed using the DESeq2 R package to detect differentially expressed genes (DEGs) with an adjusted $P$ value ($Adj.P$) less than 0.05.

## Single-cell RNA sequencing

Sequencing was carried out at the University of Texas MD Anderson Cancer Center's Advanced Technology Genomics Core (PR#002 and PDLX_PR#003) or at the NGS core facility of the Istituto Romagnolo per lo Studio dei Tumori "Dino Amadori" (PR#008). Cell viability was detected using the Countess II FL Automated Cell Counter (Thermo Fisher Scientific). Single cells were lysed, barcoded, and normalized for input onto the Chromium Single Cell A Chip Kit (10x Genomics, Pleasanton, CA, USA). Indexed sample libraries were pulled together and sequenced using a NovaSeq6000 SP 100-cycle flow cell (Illumina).

Sequencing reads were aligned to the human genome version GRCh38. A digital expression matrix was generated for each sample using Cellranger software (10x Genomics), and the data were integrated using Harmony and analyzed using Seurat (R package)[95,96]. Cells with less than 100 genes and 500 unique molecular identifiers were excluded from the analysis. Cell types were assigned using the Seurat label-transferring method and a bone marrow–specific reference (included in the SeuratData package). Conserved markers and differentially expressed genes were identified for each cluster/cell type using the Benjamini-Hochberg method to *Adj.P* for multiple testing. Statistically significant DEGs for each cluster/cell type were used to perform a gene set enrichment analysis (GSEA) using the ClusterProfiler package and the Hallmark gene-set from MSigDB. Furthermore, the identification of the leukemic cell population was assessed by computing a transcriptional signature score using the UCell R package (score ≥ 0.2) based on the expression of leukemia-associated immunophenotype for PR#002 (CD34, KIT, CD33, ANPEP, CD38, CD7, NCAM1, CD4, CD19, DNTT, ITGB3, HLA-DRA, HLA-DRB1, and HLA-DRB5), PDLX_PR#003 (CD34, CD45 KIT, CD33, ANPEP, CD38, CD2, TFRC, HLA-DRA, HLA-DRB1, HLA-DRB5, MYC, and MECOM), PR#008 in the treated and untreated samples (CD34, KIT, CD33, ANPEP, CD38, CD2, TFRC, MECOM, HLA-DRA, HLA-DRB5, HLA-DRB1, PTPRC, MYC).

### Next-generation sequencing
DNA was extracted using a Maxwell® 16 DNA purification kit (Promega, #AS1010), following the manufacturer's instructions. The concentration and purity of the DNA samples were determined with a Qubit 4 fluorometer (Thermo Fisher Scientific, #33226). Primary samples were sequenced at the NGS platform of the University-Hospital of Parma using the "Myeloid Solution" kit (Sophia Genetics SA, Saint Sulpice, Switzerland, #BS.0207.0102-48). Library preparation and sequencing were performed on a MySeq system (Illumina) following the manufacturer's instructions. Data were analyzed with Sophia DDM® software version 5.10.11.1 (Sophia Genetics SA).

### Animal Models
PDLXs were established by tail vein injection of mononucleated cells ($2.0 \times 10^6$) isolated from AML patients, PR#003, PR#008, PR#009, with t(3;3)(q21.3;q26.2), t(3;3)(q21.3;q26.2), and inv(3)(q21.3q26.2), respectively, into 6-week-old male non-obese diabetic, severe combined immune-deficient, interleukin (IL)-2 receptor gamma-deficient mice (NOD-*SCID* IL2Rgamma[null], NSG, The Jackson Laboratory, Charles River Italia). Leukemic cell engraftment was monitored every 2 weeks by flow cytometry of peripheral blood cells stained with an anti-human CD45-PE antibody (Becton Dickinson, Franklin Lakes, NJ, USA, #555483). For PDLX_PR#003, $10 \times 10^6$ cells were injected into the flank to establish a subcutaneous orthotopic model. Mice' body weight and tumor growth were evaluated every two days using caliper measurements. Tumors did not exceed 10% of the animal's body weight and had an average diameter of 15 mm. Mice were housed in specific-pathogen-free conditions (25 °C, 12-h light/12-h dark cycle, 50% humidity). Gender-based analysis was not performed because AML affects both males and females. Male mice are prioritized due to a higher incidence of AML in males compared to females. All procedures were approved under the MD Anderson (Houston, TX) Institutional Animal Care and Use Committee protocol or the N.682/2019-PR protocol at the University of Parma. Mice were sacrificed when signs of distress were observed.

For pharmacodynamic studies, three 6-week-old male NOD-*SCID* IL2Rgamma[null] (NSG, The Jackson Laboratory, Charles River Italia) mice were treated when cell engraftment of circulating hCD45 reached at least 10% of the total mononucleated cells in peripheral blood with a single administration of 10 mg/kg entinostat (PDLX_PR#003 and PDLX_PR#009) or 50 mg/kg of WS6 (PDLX_PR#009) diluted in DMSO (Sigma-Aldrich, #276855); peripheral blood cells were collected before

and after 6 hr of treatment with HDACis for immunofluorescence analysis.

For efficacy studies we used the triple-transgenic NSG-SGM3 (NSGS)(PDLX_PR#003) mice expressing human IL3, GM-CSF (CSF2), and SCF (KITLG) or NSG (PDLX_PR#008). AML leukemia cells ($1 \times 10^6$) from the PDLX_PR#003 and ($10 \times 10^6$) PDLX_PR#008 model were transplanted by tail vein injection in 12- to 16-week-old male and female NSGS mice, irradiated (125 cGy) 12 hr before transplantation (PDLX_PR#003). Engraftment of human PDLX cells was monitored every 2 weeks by flow cytometry of bone marrow–aspirated samples (PDLX_PR#003) or peripheral blood (PDLX_PR#008) stained with an anti-human CD45 antibody (BD Bioscience, Franklyn Lakes, NJ, USA, #AB11153499, #555482 and #555483, all diluted 1:40). Mice were selected for treatment randomization 4 to 6 weeks after transplantation when the human fraction of cells in the bone marrow was ≥ 0.5% or ≥0.1% in the peripheral blood. Entinostat (MedChemExpress, Sollentuna, Sweden, #HY-12163) and azacitidine (Selleck Chemicals, #S1782) were dissolved in DMSO, diluted in sterile PBS, and administered by oral gavage at 10 mg/kg per five days/week for three weeks and by intraperitoneal injection at 1 mg/kg per five days respectively. WS6 was dissolved in Tween 80 (50:50), diluted in sterile PBS, and administered by intraperitoneal injection (IP) at 25 mg/kg per five days/week for three weeks. Mice were euthanized by cervical dislocation, and organ biopsies (tibias and femurs) were collected from animals and fixed overnight in 10% neutral-buffered formalin (Sigma-Aldrich, # HT501128-4L). Tibias or vertebrae were incubated for 24 hr in Cal-Ex (Thermo Fisher Scientific, #C511-1D) to remove calcium residues and then dehydrated and preserved in 70% ethanol at ambient temperature. Fixed tissues were then embedded in paraffin following standard protocols. Leukemic cells were released from femurs and tibias by mechanic crushing in a solution of 2% FBS/PBS and filtered using 30-μm pre-separation filters (Miltenyi Biotec, Bergisch Gladbach, Germany #130-041-407) prior to analysis of hCD45 by flow cytometry.

### Statistical analysis
Statistical analyses were performed using GraphPad Prism 8 or R software. The means, standard deviation (±SD), group size, experimental details, and statistical significance are described in the figure legends in the methods section. The assumption of normal distribution was not determined, and the *P* value among samples was calculated by parametric and non-parametric *t*-test. We used one- or two-way ANOVA using statistical correction for multiple comparison algorithms, as specified in the figure legends, to determine appropriate significance among groups.

### Reporting summary
Further information on research design is available in the Nature Portfolio Reporting Summary linked to this article.

## Data availability
Publicly available dataset used in this study are available at https://www.ebi.ac.uk/biostudies/arrayexpress, https://www.ncbi.nlm.nih.gov/geo/, or https://www.cbioportal.org/ (E-GEOD 16238[19], GSE14468[31], GSE134589[32], E-MTAB-2225[15]). Data generated in this study have been deposited in ProteomeXchange Consortium PRIDE repository under accession number PXD038686 (RIME mass spectrometry), NCBI's GEO repository under accession number GSE259221 (HNT34 RNA-seq), GSE220170 (MOLM1 and UCSD/AML1 RNA-seq), GSE256129 (HNT34 ChIP-seq), GSE256130 (PR#002 scRNA-seq), GSE256040 (PDLX_PR#003 scRNA-seq), GSE256076 (PDLX_PR#008 scRNA-seq). The raw FASTQ files from NGS DNA sequencing (Sophia "Myeloid solution" panel) are available under restricted access, due to Institutial policies and privacy laws for sensitive, genomic, and personal data, in the European Genome Archive (EGA) repository at EGAD50000000506. Access for non-commercial academic use can be

granted upon an email request to the lead contact (giovanni.roti@unipr.it) within two weeks and is contingent upon a Data Access Agreement between institutions. Data will be available for six month once access has been granted. The remaining data are available within the Article, Supplementary Information or Source Data File. Source data are provided with this paper.

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

## Acknowledgements

The project was supported by an AIRC Start-up Investigator Grant (no. 17107 to G.R.), Fondazione Cariparma (3576/2017 and 0180/2018 to G.R.), Fondazione Grande Ale Onlus (to G.R.), Leukemia Research Foundation (to G.R.), Fondazione GIMEMA (to G.R) L'antica Torre di Melezzole (to M.M.), Beat-Leukemia SIES fellowship (to M.M.), Beat-Leukemia ONLUS (A.C.), Associazione Italiana contro le Leucemie-Linfomi e Mieloma, and ONLUS Parma chapter (to A.G. and G.R.) and MYNERVA project AIRC 5×1000 (to C.M. and M.P.M.). Chiara Robuschi and her family provided valuable support and inspiration throughout this work. The authors thank Enrico Maria Silini, M.D., Ph.D., for the technical support of the Pathology Department at the University of Parma, Prof. Rotraud Wieser Ph.D. from the Medical University of Vienna, who kindly shared with us the EVI1-inducible U937T model, Francesca Ruggieri and Giorgia Simonetti, Ph.D. for scRNASeq guidance, Pamela Criscuoli for administrative assistance, Roberto Ferrari, Ph.D. for ChIP-Seq supervision, Monica Crugnola M.D., Amelia Rinaldi M.D., Lucia Prezioso M.D. Ph.D., for clinical oversight, and the physicians and nurses at the Hematology and BMT unit of the University of Parma. The authors are deeply grateful to the participant of this study. The authors thank Martin Neuenschwander, Ph.D. at the Leibnitz Institute of Molecular Pharmacology and EU-OPENSCREEN Research Infrastructure Consortium (ERIC). The authors are grateful to Donna S. Neuberg, Sc.D., and Mary C. Dinauer, M.D., Ph.D., and the 2018 EHA-ASH Translational Training in Hematology faculty and scholars. Authors thank Ann Sutton for scientific proofreading at the Editing Services, Research Medical Library at the The University of Texas MD Anderson Cancer Center. The authors thank Syndax Pharmaceuticals, Inc. for providing entinostat under a compassionate use program.

## Author contributions

Conceptualization and Design, M.M., A.G., and G.R.; Methodology and Experiment Execution, A.G., E.S., M.M., A.M., N.T., F.V., C.L., E.S., A.D., C.C., C.S., A.F., V.A., B.L., F.M., A.M.L., and R.Z.; Samples Collection and Processing, E.F., B.C., G.S., L.P., M.G., G.T., S.B., A.G., E.S., E.C., A.C. M.P.M., R.L.S., and C.M.; Clinical care team, B.C., E.F., M.C., G.R., Formal Analysis L.M.D.T., S.P., A.M., C.C., M.M., and G.R.; Investigation, M.M., A.G., E.S., E.F., G.S., F.Q. and G.R.; Writing—Review & Editing, M.M., S.C., F.A., F.Q., and G.R.; Data Curation A.G., M.M., and G.R.; Supervision G.R.; All the authors read and discussed the manuscript.

## Competing interests

The authors have no competing interests.

## Additional information

[1]Department of Medicine and Surgery, University of Parma, Parma, Italy. [2]Translational Hematology and Chemogenomics Laboratory, University of Parma, Parma, Italy. [3]IRCCS Istituto Romagnolo per lo Studio dei Tumori (IRST) "Dino Amadori", Meldola, Italy. [4]Department of Leukemia, The University of Texas MD Anderson Cancer Center, Houston, TX, USA. [5]Department of Medical Science, University of Ferrara, Ferrara, Italy. [6]Hematology and BMT Unit, Azienda Ospedaliero-Universitaria di Parma, Parma, Italy. [7]Hematology and BMT Unit, Azienda USL Piacenza, Piacenza, Italy. [8]High-Throughput Screening Core Facility, CIBIO, University of Trento, Trento, Italy. [9]Computational Biology group, International Centre for Genetic Engineering and Biotechnology (ICGEB), Trieste, Italy. [10]University of Milano-Bicocca, Department of Medicine and Surgery, NANOMIB Center, Monza, Italy. [11]IRCCS Istituto Ortopedico Galeazzi, Milan, Italy. [12]Institute of Hematology and Center for Hemato-Oncology Research, University of Perugia and Santa Maria Della Misericordia Hospital, Perugia, Italy. [13]Hematology Unit, Azienda Ospedaliera-Universitaria S.ANNA, University of Ferrara, Ferrara, Italy. [14]These authors contributed equally: Andrea Gherli, Elisa Simoncini, Lucas Moron Dalla Tor. [15]Unaffiliated: Franco Aversa. ✉e-mail: giovanni.roti@unipr.it

