## [Peer Review File · Nature Communications]

Orthogonal proteogenomic analysis identifies the druggable PA2G4-MYC axis in 3q26 AMLReviewers' Comments:

Reviewer #1:

Remarks to the Author:

The editors have specifically requested I review only the mass spectrometry (RIME) related aspects of this study.

The authors have used RIME (rapid immunoprecipitation and mass spectrometry of endogenous proteins) to interrogate protein complexes in which EVI1 was the bait protein and the negative control unrelated IgG. A number of proteins were identified from three samples (2 cell lines HNT34 & UCSD/AML1 and 1 PDLX model) that were enriched over the negative controls. There are some concerns that should be addressed to improve this section of the manuscript.

(1) The raw MS (thermo QE .raw file) data is not available and without access to the PEAKS software it is difficult to evaluate the quality of the data. Could the authors please provide access to the Thermo .raw files?

(2) Could the authors detail if the RIME assays are biological replicates (at least different passages of the cell lines/different mice).

(3) The variability in RIME assays is such that typically at least 4 biological replicates are recommended to obtain statistically significant (p-values) fold changes between pull-downs. As the number of replicates for EVI1 and the corresponding IgG control was 2 in each pull down it is difficult to know the statistical confidence of the enriched proteins. Could the authors reanalyse the data using, for example, label free quantitative analysis or if possible isobaric labelling (TMT) . Ideally this would be done with a minimum of 3 biological replicates.

(4)The PDXL samples were established by tail injection of patient (AML) mononucleated cells (PR#003). More detail is required on the procedure to isolate human cells from mice for RIME analysis. This is important as there is considerable sequence identity between human and mouse proteins and contamination from mouse cells will compromise the data. For PR#003 samples are unique peptides unique for the protein or unique for human in these samples? Could the authors please detail how this sample was analysed.

(5) It is confusing what the figures in tables for each pull down refer to (eg What do the figures in columns D, H & L in table HNT34 Enriched Protein List for Evi-1 RIME Final Assay 35210 (HNT34) refer to ?

Reviewer #2:

Remarks to the Author:

The manuscript by Marchesini et al. describes the identification of HDAC inhibitors as drugs that suppress 3q26 AMLs cell viability, presumably by lowering high levels of EVI1 in these cells, and through a mechanism that it is mediated by the PA2G4-MYC axis. I am not convinced that the work as presented deserves publication in Nature Comm.

- HDAC inhibitors have already been explored as potential treatments for AML, even reaching clinical stage. The novelty in this work might be that HDAC inh might be selective and more efficacious for a subset of AML, 3q26 AMLs, because the some HDAC inh lower EVI1, a driver of cell proliferation for these cancer cells. However, although the data indicates that HDAC inhibitors indeed lower EVI1 levels in EVI1 high AML cells, the evidence is quite underwhelming that the effect seen in cell viability by HDAC inh is dependent on reducing EVI1 levels, because there is no difference in the potency of the HDAC inh tested between EVI1 high and low AML cells: Fig 2A and B show that cells TF1 and MUTZ-3 have much lower levels of EVI1 than the other AML cell lines tested, however the IC50s for cell

proliferation shown in Fig 2G are essentially the same, suggesting that EVI1 cellular levels do not matter for the potency of HDAC inh in these AML cells. In addition, Fig 3E shows AUC data for the effect of AR-42, belinostat, entinostat for 5 EVI1High and 7 EVI1Low AML cell lines and the actual differences are very small, although somewhat statistically significant. Same for patient derived cells in Fig 4C. In addition, Fig S1 D shows that, as a class, HDAC inh are not selective for EVI1 high cells. In general, the authors do not show convincing data that the pharmacological effects of HDAC inhibitors on cell viability are larger for AVI1 high vs low cells, and draw a clear conclusion that the effects seen in cell growth are due to AVI1 downregulation.

- The discovery that a PA2G4 inhibitor downregulates AVI1 appears to be novel, and in fact, the pharmacological effects on cell viability shown for a PA2G4 inhibitor appears to be more dependent on AVI1 levels (Fig S9 A) than HDAC inh. Puzzling that the authors never tested the combination of HDAC inh and the PA2G4 inhibitor to test whether there might be some additive or synergistic effects.

- It is unclear what the p-values in Table S1-1 headers refer to; the screen was done in duplicate.

What groups are compared to calculate p-values from the screening data? The Δ POC index includes a differences in % cell viability to control and an Adj.P. How is this Adj.P calculated?

- Line 100: "The HDACis AR-42, belinostat, trichostatin A, and entinostat preferentially affect the proliferation of EVI1High AML cells compared to EVI1low". What selectivity criteria was used to identify selective EVI1 high vs low compounds? Did the criteria include both potency (IC50) and efficacy (% cell viability).

- Line 110: "Most (56.2%) of the HDACis sampled in the CMap libraries scored among the top 250 hits (P = 0.0006)". I am assuming this statement refers to the cell proliferation screen, so I am then confused as to what is plotted in Fig 1E. In addition, there is no mention in the text to the panels for the other compound classes. Shouldn't the y-axis be % cell viability in Fig 1E?. The legend seems to suggest that the x-axis is number of perturbagenes in the connectivity map. It is not clear how many total number of perturbagenes in the connectivity map: 2836 total or 2836 small molecules? From the x-axis, it might seem that there are >4000 perturbagenes total.

- Line 158: The IC50 values have a high number of significant digits. Are their measurements really that precise? What are the errors for the IC50 values?. In this regard, the authors should include the number of biological and technical replicates whenever they show error bars, eg. Fig 2G.

- I don't understand Fig 1F. What is the p-value? Is it an enrichment score for the % of compounds in each target class that were considered active (top 90% by Δ POC index) in the cell viability screen?

- In Fig5G and H, there is no comparison of MYC levels in AVI1 high vs low cells. Is there a correlation between the levels of both proteins in AML cancer cells? that would help demonstrate that the pharmacological effect of HDAC inh on MYC levels in these AVI1 high cells is through downregulation of AVI1 and not directly on transcription regulation of MYC bu HDAC inh, which has been shown before.

- Also, there is not data shown for the effects of WS6 inhibited cell viability in AML EVI1 High vs low cell lines and clinical samples to demonstrate that the pharmacological effects of PA2G4 inh on 3q26 AMLs cell viability is indeed EVI1 dependent.

Reviewer #3:

Remarks to the Author:

In this study Dr. Marchesini and co-authors used integrated high-throughput drug screening and expression profile analysis to identify selective inhibitors of EV1/MECOM signaling in AML with 3q26 abnormalities, one of the AML subtypes with the most dismal outcome and lacking targeted and effective therapies. HDACIs were demonstrated potent repressors of EVI1 and further examined in downstream analyses.

Major comments

Page 5, line 87: the authors should specify that the MOLM1 cell line used for the initial screen, although carrying an inv(3) leading to MECOM (EVI1) overexpression, is a chronic myeloid leukemia

and not acute myeloid leukemia.

In order to find candidates for effective inhibition in AML cells with EVI1 deregulation, the authors intersected the results from the high-throughput drug screening with those from the gene expression differential analysis of TF-1 cells with siRNA targeting EVI1. However, only one cell line, TF-1, not included in the drug screening, was analyzed. An alternative approach may be to knock-down EVI1 in MOLM1 or UCSD-AML1 cells, perform transcriptome sequencing and then compare the gene expression data with the results from the drug screening and from the experiment in TF1 cells. Moreover, in order to strengthen their results, the authors could also analyze the public available data from the Cancer Dependency Map in AML cell lines with EVI1 overexpression and define a common list of targets.

Figure 2F: did the authors analyze immunophenotype in the three conditions (NT, EVI1 sh#16 and EVI1 sh#87)?

Page 16 line 318: I think the authors meant "HDACIs" instead of "HDAC".

The authors hypothesized that HDACIs can impair the function of the EVI1 co-transcriptional complex and profiled by RNAseq two AML cell lines with high expression of EVI1 post 16 hours treatment with AR-42 and entinostat and found a strong association with MYC signature genes. One key experiment that is missing to demonstrate the deregulation of the EVI1 transcriptional complex is the analysis of EVI1 chromatin-binding together with H3K27ac profiling by ChIP-seq (or CUT&RUN) in cells pre and post treatment.

The scRNAseq analysis on follow-up samples provides interesting findings on how the combination azacitidine and entinostat targets stem and progenitor cells, however, as the authors are aware, it does not distinguish the effect of the single drugs. Can the authors perform cell viability assay and single cell RNAseq on normal CD34+ overexpressing EVI1 (for example by viral transduction) or in PDX cells before and after treatment with the combination and single azacitidine and entinostat?

We would like to express our appreciation for the thoughtful comments provided by the Reviewers. Their feedback has influenced our study, guiding us in addressing important aspects and refining our methods and interpretations.

In response to their feedback, we have prepared a point-by-point rebuttal, which will be presented below. We would like to acknowledge that we have created n=35 figures panels, intended to clarify specific points raised by the Reviewers. However, upon careful consideration, we decided to include only part of these data in the revised version of the manuscript. The remaining data are presented here as part of our rebuttal.

Reviewer #1:

The editors have specifically requested I review only the mass spectrometry (RIME) related aspects of this study.

The authors have used RIME (rapid immunoprecipitation and mass spectrometry of endogenous proteins) to interrogate protein complexes in which EVI1 was the bait protein and the negative control unrelated IgG. A number of proteins were identified from three samples (2 cell lines HNT34 & UCSD/AML1 and 1 PDLX model) that were enriched over the negative controls. There are some concerns that should be addressed to improve this section of the manuscript.

(1) The raw MS (thermo QE .raw file) data is not available and without access to the PEAKS software it is difficult to evaluate the quality of the data. Could the authors please provide access to the Thermo .raw files?

We are pleased to inform you that we can provide access to the Thermo.raw files at:

ProteomeXchange title: RIME of EVI1 in HNT34 and UCSD/AML1 cell lines and one primary sample to identify interactors

ProteomeXchange accession: PXD042440

PubMed ID: Not applicable

Publication DOI: Not applicable

Project Webpage: <http://www.ebi.ac.uk/pride/archive/projects/PXD042440>

FTP Download: <ftp://ftp.pride.ebi.ac.uk/pride/data/archive/2023/05/PXD042440>

(2) Could the authors detail if the RIME assays are biological replicates (at least different passages of the cell lines/different mice).

In this work we wanted to explore the native interactors of EVI1 by leveraging the RIME assay. For this purpose, we used the HNT34 and UCSD/AML1 cell lines, as well as the clinical PR#003 sample. Hence, for this experiment, the biological replicates representative of a 3q26 AML model are equal to three (n=3). All these models exhibited the t(3:3)(q21;q26) genomic rearrangement, and each experiment was carried out in technical replicates (n=2).

(3) The variability in RIME assays is such that typically at least 4 biological replicates are recommended to obtain statistically significant (p-values) fold changes between pull-downs. As the number of replicates for EVI1 and the corresponding IgG control was 2 in each pull down it is difficult to know the statistical confidence of the enriched proteins. Could the authors reanalyse the data using, for example, label free quantitative analysis or if possible isobaric labelling (TMT). Ideally this would be done with a minimum of 3 biological replicates.

We thank the Reviewer for bringing attention to this aspect as it gives us the opportunity to better clarify our data analysis workflow.

RIME assays have been successfully used in various research fields to study protein interactions, functions, and disease mechanisms^{1, 2, 3, 4, 5, 6, 7, 8, 9, 10}. The data obtained from these assays, even with a limited number of replicates, have helped to depict the protein interactome and potential biological pathways involved in different cancer models^{11, 12, 13}.

In our analysis, we focused on proteins that were present in the pull-down sample but absent in the IgG control. This is consistent with the literature in this field.

In Supplementary **Table S4-1**, we report the lists of uniquely identified proteins for all samples after removing proteins present in the IgG control and other proteins considered as background (below the spectral count of 5). This list was generated by averaging the spectral counts for common elements as identified in replicate R1 and R2 pull-downs for each model. The averaged spectral counts between biological experiments are also reported in the table (**column E**). Excluding the identification of the bait protein EVI1, a total of 155, 107, and 117 proteins were identified in PR#003, HNT34, and UCSD/AML1 samples, respectively (**Figure R1A**). Then, only proteins that were common to all three biological replicates were considered for further investigation. This resulted in 68 common targets, excluding EVI1, as reported in **Table S4-1** and **Figure S9C**. The description of **Table S4-1** that has been partially revised in this new version of the manuscript.

Hence, in our original draft, we did not perform a fold change analysis, and it was not our intent to do so. The goal was consistently to identify a set of EVI1 interactors using methodological approaches that have been successfully explored in the literature.

However, as suggested by the reviewer, we have reanalyzed our data using a label-free quantification (LFQ) method. In **Figure R1B**, we report the correlation analyses based on the protein LFQ intensities between biological and technical replicates. We observed a robust correlation of LFQ signals among both technical ($r = 0.9493$ PR#003; $r = 0.9668$ UCSD/AML1 and $r = 0.9533$ HNT34) and biological ($r = 0.8981$ HNT34-UCSD/AML1; $r = 0.8548$ HNT34-PR#003 and $r = 0.8276$ UCSD/AML1-PR#003) replicates, indicating a highly reproducible, relative LFQ between samples.

For a quantitative analysis, we included our technical and biological replicates ($n = 6$) in a group named "3q26 AML" and compared them to the group including all the correlative IgG controls ($n=6$), named "Control". The Thermo raw files were analyzed using MaxQuant (MQ) version 2.4.2.0. The LFQ intensities of proteins from MQ analysis were imported and filtered for reverse identifications (false positives), contaminants, and proteins "only identified by site". Data were transformed to \log_2 scale. Then, we imputed missing values and replaced them from a normal distribution. The protein quantification and calculation of statistical significance were performed with a two-sample t-test and with a permutation-based correction controlled by using an FDR threshold of 0.05. A protein was considered as EVI1's interactor if the difference between the "3q26 AML" and "Control" groups was statistically significant ($P < 0.05$), the fold change was ≥ 4 , and it was identified with a minimum of two peptides. Using this approach, we identified 147 proteins that were enriched in 3q26 AML samples with a fold change greater than 4 (**Table S4-4**). We then asked whether the group of EVI1-MYC interactors identified by combining RIME and ssGSEA (as described in the manuscript and presented in **Figure S9D** and **Table S4**) were also identified using this alternative MS data analysis. As shown in **Figure R1C** and presented in this new version of the manuscript in **Figure S9E**, we found that 10 out of 11 interactors scored in the LFQ

analysis (RACK1, scored as an EVI1 interactor by RIME, was not confirmed by this analysis). Additionally, we observed that, as in the case of the RIME analysis, EVI1 co-regulators identified previously using isotope labeling-based MS¹⁴ were represented among the 3q26 AML enriched proteins (**Figure R1C**).

To further validate the results obtained both by RIME and LFQ analysis, we performed biochemical co-immune precipitation (co-IP) experiments to confirm EVI1 and PA2G4 binding. We chose the HNT34 cell line as a 3q26 AML model based on the relatively higher expression level of EVI1 (**Figure 2A** and **4D**) in the effort to compensate for the difference in sensitivity between co-IP and RIME. We observed a reciprocal co-immunoprecipitation of EVI1 and PA2G4. We repeated a co-IP in 293T cells after an ectopic overexpression of both *EVI1* (NM_005241.3) and *PA2G4* (Gene BankBC001951.1) open reading frames and confirmed, also in this setting, a reciprocal co-IP between EVI1 and PA2G4 (**Figure R1D**, included in the manuscript, **Figure S9G**).

Despite we are aware that RIME is an exploratory experimental approach, the selection of PA2G4 was based on multiple orthogonal evidences, including its potential significance in the context of 3q26 leukemia and its modulation by HDAC inhibitors.

- First, PA2G4 was identified as a common protein across three 3q26 leukemia models, including primary-derived cells. We provided evidence of a biochemical interaction between EVI1 and PA2G4 interaction that corroborates co-localization studies (**Figure S9F**). In addition, as reported in **Table S4-2** of our manuscript PA2G4 was part of a long list of EVI1-interactors in different disease models (n=236) of wild type *EVI1* transduced 293T¹⁵.
- Second, *PA2G4* is part of *MYC*-dependent signatures, and both the Myc pathway and PA2G4 were modulated by HDAC inhibitors in 3q26 AML models.
- Third, overexpression of *PA2G4* rescued 3q26 AML from the HDACi-induced phenotype, suggesting its involvement in the mechanism of action of these molecules.

In conclusion, we believe that RIME assay, despite its innate limitations that are similar to other approaches, provides valuable insights into the protein interactome and potential biological mechanisms given the intersectionality of this approach with the rest of the study. We will continue to refine our methods and explore opportunities to incorporate alternative analyses in future studies.

(4) The PDXL samples were established by tail injection of patient (AML) mononucleated cells (PR#003). More detail is required on the procedure to isolate human cells from mice for RIME analysis. This is important as there is considerable sequence identity between human and mouse proteins and contamination from mouse cells will compromise the data. For PR#003 samples are unique peptides unique for the protein or unique for humans in these samples? Could the authors please detail how this sample was analysed.

We thank the Reviewer for this comment and concern. RIME analysis was carried out referring to human protein database. In fact, PDXL samples (PR#003) used in this specific context were established by tail injection of patient leukemic cells into mice. Subsequently, the cells were transplanted subcutaneously to ensure an adequate supply of protein for analysis. These steps were taken to maximize the representation of human proteins in the samples and to allow for sufficient protein production. The transplanted tumor maintained the original translocation, as shown by FISH analysis.

We apologize if this information was not explicitly stated in the text, but it was presented in **Figure S7A** and **S7B** of original version of the manuscript (now in **Figure S9A-B**) where the experimental procedure and analysis workflow are described in more detail.

(5) It is confusing what the figures in tables for each pull down refer to (eg What do the figures in columns D, H & L in table HNT34 Enriched Protein List for Evi-1 RIME Final Assay 35210 (HNT34) refer to?

We apologize for the missing information. The file mentioned by the Reviewer, sent during the revision, contains the lists of uniquely identified proteins in each RIME experiment (HNT34 sample in this case). Proteins considered as background (spectral count < 5) and proteins found in the negative control (IgG) were previously filtered. Lists going from columns B to D and J to L were obtained by overlapping candidate peptides identified in the replicates R1 (B to D) and R2 (J to L) and their correlative IgG negative controls. The list going from F to H contains proteins that were identified in both replicates. Gene names (columns B, F and J) and protein names (columns C, G and K) are indicated. Columns D and L accommodate the spectral counts of uniquely identified proteins (with a spectral count > 5) of the replicates R1 and R2 respectively. The column H contains instead the averaged spectral count for common elements as identified in R1 and R2 pull-downs.

Reviewer #2:

The manuscript by Marchesini et al. describes the identification of HDAC inhibitors as drugs that suppress 3q26 AMLs cell viability, presumably by lowering high levels of EVI1 in these cells, and through a mechanism that it is mediated by the PA2G4-MYC axis.

HDAC inhibitors have already been explored as potential treatments for AML, even reaching clinical stage. The novelty in this work might be that HDAC inh might be selective and more efficacious for a subset of AML, 3q26 AMLs, because the some HDAC inh lower EVI1, a driver of cell proliferation for these cancer cells. However, although the data indicates that HDAC inhibitors indeed lower EVI1 levels in EVI1 high AML cells, the evidence is quite underwhelming that the effect seen in cell viability by HDAC inh is dependent on reducing EVI1 levels, because there is no difference in the potency of the HDAC inh tested between EVI1 high and low AML cells: Fig 2A and B show that cells TF1 and MUTZ-3 have much lower levels of EVI1 than the other AML cell lines tested, however the IC50s for cell proliferation shown in Fig 2G are essentially the same, suggesting that EVI1 cellular levels do not matter for the potency of HDAC inh in these AML cells.

We understand the concern of the Reviewer, however we would like to point out that clinical and preclinical studies have demonstrated that different subtypes of AML may respond differently to specific HDAC inhibitors (HDACis) based on the presence, for example, of leukemia-associated fusion proteins (LAFPs) or specific genetic alterations rather than solely relying on the expression level of the specific target^{16, 17}.

For instance, studies have shown that certain LAFPs, such as *PLZF::RAR α* , *PML::RAR α* , and *RUNX1::RUNX1*^{16, 17, 18} can sensitize AML cells to HDAC inhibition, leading to improved response rates with specific HDACis. This suggests that the recruitment of HDACs to specific promoters through the interaction of fusion proteins can enhance the susceptibility of cells to HDAC inhibition. Additionally, the response to different HDACis can be influenced by specific molecular pathways. Belinostat promotes an anti-leukemia effect in several AML cell lines, especially in acute promyelocytic leukemia and, in combination

with all-trans retinoic acid, accelerates granulocytic differentiation¹⁹. However, in this case, the mechanism that explains why the chromosomal translocation t(15;17) bearing *PML::RARα* fusion differs in susceptibility to HDACis has not been investigated. Entinostat instead degrades FLT3 by inhibiting the chaperone protein 90 in AML cells²⁰ and reduces *MYC* transcription in *PICALM::MLLT10* or *PML::RARα* rearranged AML cell lines²¹. Unfortunately, in phase I²² or phase II study of entinostat^{22, 23}, with and without azacitidine, patients were not segregated by cytogenetic mutational status, limiting any potential conclusion of entinostat activity on specific AML subgroups. This appears to be a more general limitation for AML trials.

It is important to note that the understanding of the precise mechanisms underlying the differential responses to HDACis in various AML subgroups, including those with specific chromosomal translocations or genetic alterations like *EVI1*, is still limited.

That said, it is absolutely possible that other mechanisms are involved in cell-line specific response to HDACis. However, in this version of manuscript we further provided evidence that the expression of *EVI1* sensitizes cells to HDAC inhibition (**Figure R2 A-H**).

These new data are presented in **Figures 3E-I, S4E** and **S8C-D** in the new version of the manuscript. Furthermore, we included an additional 17 primary cases to our study (**Figure 4C**) that confirmed our original results.

In addition, Fig 3E shows AUC data for the effect of AR-42, belinostat, entinostat for 5 *EVI1*^{High} and 7 *EVI1*^{Low} AML cell lines and the actual differences are very small, although somewhat statistically significant. Same for patient derived cells in Fig 4C.

The small differences between *EVI1*^{Low} and *EVI1*^{High} are somewhat expected, considering the potency of HDACis in several tumor models including AML. We acknowledge that the observed differences in drug response may be influenced by multiple mechanisms driven by HDACs, potentially resulting in varying responses even within the *EVI1*^{Low} group. For example, we have demonstrated the presence of two distinct populations in MOLM1 cells (we excluded contamination by repeating this observation in multiple STR profiled batches): one that is *EVI1*-positive, characterized by large nuclei and prominent nucleoli, and another consisting of *EVI1*-negative cells, which are smaller in size and exhibit weak or no *EVI1* protein expression (**Figures 2B, 3G-H**). We sorted MOLM1 cells based on this characteristic and subsequently tested HDAC inhibitors. In this isogenic model, HDAC inhibitors displayed greater efficacy in MOLM1-*EVI1*^{High} cells compared to MOLM1-*EVI1*^{Low} cells, which, at certain concentrations, still exhibited sensitivity to HDAC inhibition (**Figure 3I**).

The inclusion of additional data, further described below, in **Figures 3E-I, S4E, S8C-D** and **Figure 4C**) and the demonstration of differences in drug response between *EVI1*^{High} and *EVI1*^{Low} populations further support our findings. These results suggest that there may be distinct molecular characteristics and sensitivities to HDACis in the *EVI1*^{High} subgroup, which can contribute to the observed differences in drug response.

In addition, Fig S1 D shows that, as a class, HDAC inh are not selective for *EVI1* high cells. In general, the authors do not show convincing data that the pharmacological effects of HDAC inhibitors on cell viability are larger for *EVI1* high vs low cells and draw a clear conclusion that the effects seen in cell growth are due to *EVI1* downregulation.

We appreciate the Reviewer's input but respectfully disagree with the statement that our data does not demonstrate the differential effects of HDAC inhibitors on *EVI1*^{High} vs. *EVI1*^{Low} cells or support the conclusion that the observed effects on cell growth are due to *EVI1*

downregulation. First, we have shown that the absence of EVI1 leads to a significant delay in cell growth in 3q26 AML, indicating its role in promoting cell proliferation. In this version of the manuscript we included a second *EVI1* silenced 3q26 AML cell lines (**Figure R3A-C** and **Figure S3A-C** of the manuscript).

Secondly, HDAC inhibitors are known to have diverse effects²⁴ and their response is not expected to be a simple binary "yes" or "no" outcome, as seen with certain targeted tyrosine kinase inhibitors. We understand that the data presented in **Figure S1D** may not show a perfect segregation between EVI1^{High} and EVI1^{Low} cells in terms of their response to HDAC inhibitors. In **Figure S1D**, we have indeed tested both selective and pan-HDAC inhibitors of different classes (e.g selective HDAC inhibitors pyroxamide, tacedinaline, ISOX, WT-161, entinostat, trichostatin-a, Merk60, mocetinostat, vorinostat, NCH-51, apicidin, droxinostat and others; pan-HDAC inhibitors panabinstat, HC toxin, givinostat, belinostat, phenylbutyrate, dacinostat and others). The variability in selectivity among HDAC inhibitors can indeed contribute to the observed differences in the degree of the effect. The diverse selectivity profiles of HDAC inhibitors can result in differential modulation of histone and non-histone proteins, leading to distinct cellular responses. Some inhibitors may primarily target specific HDAC isoforms associated with critical pathways in the context of EVI1 expression and 3q26 status, thereby resulting in a more pronounced effect. Nevertheless, it is important to consider that the responses observed are relative and clearly demonstrate a trend towards enhanced sensitivity in EVI1^{High} cells. We discussed this aspect more explicitly in the revised manuscript and in the answers above.

To meet the Reviewer' expectations, we have indeed included two new experimental models to investigate the role of EVI1 and the activity of HDAC inhibitors. As described above, we isolated MOLM1-EVI1^{High} cells from MOLM1 cells and showed their greater dependency on the HDAC machinery compared to MOLM1-EVI1^{Low} cells (**Figure 3G-I**). We overexpressed *EVI1* in HL60 cells and demonstrated that EVI1-cells are more sensitive to HDAC inhibitors compared to non-expressing cells (**Figure 3E-F** and **S4E**). Finally, we have included additional new primary cases that collectively recapitulate the data presented in the original submission (**Figure 4C** and **Table S2**).

In summary, we conducted five different types of comparisons, involving AML cell lines, inducible models (U937T), stable overexpression (HL60), isogenic models (MOLM1^{EVI-high} vs. MOLM1-EVI1^{Low} and primary samples. These comparisons consistently indicate that the 3q26 status/EVI1 expression sensitizes cells to HDAC inhibition, resulting in a reduction in EVI1 levels. Furthermore, this outcome is, in part, mediated by PA2G4 (**Figure 7B**). Importantly, we observed the same preferential activity toward EVI1^{High} cells with a PA2G4 inhibitor, suggesting a potential avenue for drug development.

The discovery that a PA2G4 inhibitor downregulates AVI1 appears to be novel, and in fact, the pharmacological effects on cell viability shown for a PA2G4 inhibitor appears to be more dependent on AVI1 levels (Fig S9 A) than HDAC inh. Puzzling that the authors never tested the combination of HDAC inh and the PA2G4 inhibitor to test whether there might be some additive or synergistic effects.

We appreciate the reviewer's recognition of the novelty in our discovery of a PA2G4 inhibitor downregulating EVI1. We have indeed conducted the suggested experiment and evaluated the combination of HDAC inhibitors and the PA2G4 inhibitor. The results demonstrate an additive/synergistic effect on cell viability when combining HDAC inhibitors with the PA2G4 inhibitor WS6. These findings support the idea that PA2G4 may indeed act as a mediator of

the HDAC response. These findings are shown in **Figure R3D**, and included as **Figure S11B** in the new version of the manuscript.

It is unclear what the p-values in Table S1-1 headers refer to; the screen was done in duplicate. What groups are compared to calculate p-values from the screening data? The Δ POC index includes a differences in % cell viability to control and an Adj.P. How is this Adj.P calculated?

We apologize to the Reviewer for not being clearer about the screening procedure in our methods section and Table S1-1. We have revised this section in the new version of the manuscript, as suggested by the Reviewer.

The screening was performed in duplicate, and the values in column C (POC_molecules) of **Table S1-1** refer to the average POCs derived from technical replicates. The POC_molecules values for each compound were compared to the DMSO control values (n = 24) using an ANOVA test applied to a linear regression model. The resulting p-values were adjusted for multiple comparisons using the Benjamini & Hochberg (BH or FDR) method. We also ran a one-sample t-test to corroborate this analysis. This test is more appropriate for comparing a single value (POC_molecules) to a series of data (DMSO n = 24). As in the case of the ANOVA test, we adjusted the resulting p-values using BH correction. As reported in **Table R1**, the results confirmed what observed with the previous statistical test, thus we decided to not include the one sample t-test in the final version of our manuscript.

Line 100: "The HDACs AR-42, belinostat, trichostatin A, and entinostat preferentially affect the proliferation of EVI1^{High} AML cells compared to EVI1^{low}". What selectivity criteria was used to identify selective EVI1 high vs low compounds? Did the criteria include both potency (IC50) and efficacy (% cell viability).

To compare the effect of single agents between EVI1^{High} and EVI1^{Low} models (**Figure S1D**) we choose the Area Under the fitted dose response Curve (AUC) criterion. AUC improves the predictive accuracy for classifying samples as sensitive or resistant compared to more traditional metrics such as IC50, or % cell viability. A substantial body of literature has concluded that AUC, a parameter that combines potency and efficacy into a single measure^{25, 26}, is robust approach and surpasses response metrics when the goal is to compare a single drug across cell lines exposed to identical dose ranges.

For example, IC50 presumes a typical sigmoidal shape in dose-response curves, with there's no growth inhibition in the absence of the compound and complete growth inhibition at high compound doses. This assumption doesn't distinguish between samples that reach 50% growth inhibition at the same dose, even if one of the samples achieves significantly greater growth inhibition at higher doses. Consequently, we consistently used AUC in experiments requiring the categorization compounds sensitive or resistant in EVI1^{High} and EVI1^{Low} in cell lines (**Figure 3J, S8E**) or primary samples (**Figure 4C and R3H**, included as **7B** in the manuscript) and in MYC^{High} and MYC^{Low} **Figure R3F-G (S8B, S8F** of the new version of the manuscript).

- Line 110: "Most (56.2%) of the HDACs sampled in the CMap libraries scored among the top 250 hits (P = 0.0006)". I am assuming this statement refers to the cell proliferation screen, so I am then confused as to what is plotted in Fig 1E. In addition, there is no mention in the text to the panels for the other compound classes. Shouldn't the y-axis be % cell viability in Fig 1E?. The legend seems to suggest that the x-axis is number of perturbagens in the connectivity map. It is not clear how many total number of perturbagens in the

connectivity map: 2836 total or 2836 small molecules? From the x-axis, it might seem that there are >4000 perturbagenes total.

A connectivity mapping approach was used to prioritize small molecules as previously described²⁷. Top 100 up- and down-regulated probes based by the signal-to-noise ratio, defining an *EV11* “On” versus an “Off” change from the GSE16238 dataset were submitted to the cMap-Touchstone query version 1.1 (<https://clue.io/query>) of 8864 perturbagens, of these 2836 are compounds. In CMap, the summary score is a perturbagen-centric measure of connectivity that summarizes the results observed in individual cell types, while the connectivity score (in our model *EV11* “Off” score) is a quality control measurement comparing an observed enrichment score from *EV11* gene set to reference gene sets, ranging from -100 to 100 and, for perturbagenes with a positive connectivity score > 0 (n= 4187), presented as $\text{Log}_{10}(2^{\text{score}})$ (= highest correlation with an *EV11* “Off” state). In this regard, the y axis is correct since it refers to a compound's probability of mimicking an *EV11* “Off” state. The x axis indicates n= perturbagenes. We clarified this point in the method section and in the figure legends.

- Line 158: The IC50 values have a high number of significant digits. Are their measurements really that precise? What are the errors for the IC50 values? In this regard, the authors should include the number of biological and technical replicates whenever they show error bars, eg. Fig 2G.

We used a digital dispenser TECAN D300e for small molecules dispensing, and a peristaltic cells dispenser for *in vitro* assays ensuring an accurate measurement of the IC50 and AUC values. The accuracy of the D300e is specified as $\pm 0.2\%$ of the dispense volume, with a dead volume of 2 μL . This means that the D300e can dispense volumes of liquid with an accuracy of $\pm 0.02 \mu\text{L}$. This information was mentioned in the methods section of the manuscript. Additionally, in this version of the manuscript, we have incorporated the standard deviation to indicate the range of small molecule activity, providing a measure of variability in our measurements. The number of biological and technical replicates was present in the figure legends of the original version of the manuscript. We have included the error for IC50 value in the manuscript.

I don't understand Fig 1F. What is the p-value? Is it an enrichment score for the % of compounds in each target class that were considered active (top 90% by ΔPOC index) in the cell viability screen?

The analysis presented in **Figure 1F** of the submitted version of our manuscript is now the **Figure S2B** panel and derives from the results obtained from the *in silico* small molecule screening. The goal was to determine if the cMap enrichment of HDACi or histamine, glucocorticoid, and acetylcholine receptor antagonists (as examples of groups of molecules in the library with comparable or higher size than HDACi). To check if the rank of these classes of molecules was specific or not, we asked if it could be statistically significant with respect to random sets. To address this challenge, we created random sets of the same size of the class of molecule of interest (n = 32 in the case of HDACi, n = 52 for histamine receptor antagonists, n = 47 for glucocorticoid receptor antagonists and n = 66 for acetylcholine receptor antagonists). Based on the fact that the generated artificial family of molecules had undefined distributions, we leveraged a statistical approach based on the parametric bootstrap method²⁸ applied to gene-set analysis as described in the study of Coombes et al.²⁹ obviously referred in our case to molecule-sets. This method allows us to

estimate the uncertainty of a p-value. We obtained rank values derived from our random sets, which we compared to the one of interest, using the Wilcoxon test. By following the bootstrap approach, we averaged these results to obtain a p-value whose stability was checked by repeating this procedure 100 times.

We did not adjust the p-values for multiple comparisons for the following reasons:

- The results could have been influenced by the high number of tests, and thus self-defeating to the aim of evaluating p-values robustness and stability.
- The multiple comparisons had a common data set (rank of molecule of interest), and thus the analysis was not based on independent measurements, a mandatory requirement of the p-value adjustment procedure.

Therefore, **Figure S2B** shows the distribution of p-values obtained by independent runs of the method described above for the indicated groups of molecules.

In Fig5G and H, there is no comparison of MYC levels in AVI1 high vs low cells. Is there a correlation between the levels of both proteins in AML cancer cells? that would help demonstrate that the pharmacological effect of HDAC inh on MYC levels in these AVI1 high cells is through downregulation of AVI1 and not directly on transcription regulation of MYC bu HDAC inh, which has been shown before.

We performed the experiment suggested by the reviewer and quantified the levels of MYC in the AML cell line available in the laboratory. This data (**Figure R3E**) is now presented in **Figure S8A**. Overall, the data show different MYC levels in 3q26 AML, with some cell lines representing clear exceptions. If MYC was the primary target of the HDAC dependency, one would have expected greater activity in MYC^{High} cells. However, this is not the case, **Figure S8B (Figure R3F)** but, as expected, it is for JQ1 an inhibitor of MYC superenhancers (**Figure R3G or S8F** in the new version of the manuscript). Instead, when we distinguish the response to HDACis based on the 3q26 status/EVI1^{High} groups, there is indeed statistical significance as shown in **Figure 3J** and **Figure 4C**. Furthermore, conditional expression of EVI1 leads to an enhanced anti-proliferative effect of HDACis, which is not the case when MYC is overexpressed.

Moreover, genetic suppression of EVI1 leads to a reduction in MYC levels, suggesting that the loss of EVI1 contributes to the decrease in MYC in the 3q26 model.

Also, there is not data shown for the effects of WS6 inhibited cell viability in AML EVI1 High vs low cell lines and clinical samples to demonstrate that the pharmacological effects of PA2G4 inh on 3q26 AMLs cell viability is indeed EVI1 dependent.

We appreciate the Reviewer's feedback on this topic. In response, we assessed the effects of WS6 on cell viability in AML primary samples with varying levels of EVI1 expression (EVI1^{High} n= 8 or EVI1^{Low}, n= 15) as shown in the **R3H**, included as **Figure 7B** of the new version of the manuscript.

The results obtained from these experiments support the idea of a dependence of 3q26 AML on the PA2G4 protein. These results also indicate that further optimization of the PA2G4 protein could potentially lead to the development of drug-like molecules.

Reviewer #4:

In this study Dr. Marchesini and co-authors used integrated high-throughput drug screening and expression profile analysis to identify selective inhibitors of EV1/MECOM signaling in AML with 3q26 abnormalities, one of the AML subtypes with the most dismal outcome and

lacking targeted and effective therapies. HDACi were demonstrated potent repressors of EVI1 and further examined in downstream analyses.

Major comments

Page 5, line 87: the authors should specify that the MOLM1 cell line used for the initial screen, although carrying an inv(3) leading to MECOM (EVI1) overexpression, is a chronic myeloid leukemia and not acute myeloid leukemia.

In this version of the manuscript, we have accurately defined the MOLM1 cell line as being established from a blastic transformation of chronic myeloid leukemia (CML). We have further substantiated this characterization by conducting a karyotype analysis (**Figure R3I**), which confirmed the presence of a complex karyotype, as previously reported in ³⁰."

In order to find candidates for effective inhibition in AML cells with EVI1 deregulation, the authors intersected the results from the high-throughput drug screening with those from the gene expression differential analysis of TF-1 cells with siRNA targeting EVI1. However, only one cell line, TF-1, not included in the drug screening, was analyzed. *An alternative approach may be to knock-down EVI1 in MOLM1 or UCSD-AML1 cells, perform transcriptome sequencing and then compare the gene expression data with the results from the drug screening and from the experiment in TF1 cells.* Moreover, in order to strengthen their results, the authors could also analyze the public available data from the Cancer Dependency Map in AML cell lines with EVI1 overexpression and define a common list of targets.

We appreciate the reviewers' valuable suggestions. One of the reasons we chose TF1 cells is their high transduction efficiency compared to UCSD1/AML1 and MOLM1 cell lines. We encountered significant resistance to lentiviral transduction in 3q26 AML cell lines, including unsuccessful attempts with MOLM1 cells. This difficulty in transducing 3q26 AML cell lines likely explains the absence of such cell lines in the Cancer Dependency Map, except for TF1 cells. Consequently, generating a comprehensive list of putative targets from the Cancer Dependency Map becomes unrealistic due to the limited representation of EVI1^{High} cases. To overcome this limitation, we used a transcriptional dataset (E-MTAB-2225) ³¹. of AML and characterized EVI1^{High} and EVI1^{Low} profiles. Through the analysis of transcriptional profilings, we established a signature that distinguishes between 3q26-positive and 3q26-negative cell lines as shown in **Figure R4A (Figure S2F** of the new version of the manuscript). Subsequently, as shown in **Figure R4D-E** we used the LINCS database as a complementary approach to cMAP and identified HDAC inhibitors (HDACi) as a compound class associated with the transition from EVI1 "On" to EVI1 "Off" state (data included in **Figure 1F and S2G** of the manuscript). Furthermore, we successfully sequenced the HNT34 cell line in which *EVI1* was lentivirally silenced, as demonstrated in **Figure S3A-C**, and revealed a significant overlap between EVI1-regulated gene signatures in HNT34 and EVI1-dependent genes in TF1 (**Figure R4C**, included in the manuscript as **Figure S2E**) indicating a consistent signature derived from EVI1 modulation in these models. Strikingly, the projection of the HNT34 signature onto the LINCS L1000 dataset space demonstrated that transcriptional changes associated with an EVI1 "Off" status mimic and match HDACi signatures as shown in **Figure R4D-E** included in the new version of the manuscript (**Figure 1F and S2G**).

Furthermore, to validate the results of the *in silico* small molecule screening in TF-1 cells (**Figure 1D-E and S2B**), we performed a tertiary small molecule screen of 4942 small drugs that confirmed the potent antiproliferative activity of HDACi compared to non the *EVI1* "Off"

inducers histamine, glucocorticoid, acetylcholine receptor antagonists or to other antineoplastic compounds contained in Selleck and Sigma-Aldrich collection screened as shown in **Figure R4F**. We presented this new results in **Figure S2C** and **Table S1** of this new version of the manuscript.

Figure 2F: did the authors analyze immunophenotype in the three conditions (NT, *EVI1* sh#16 and *EVI1* sh#87)?

In **Figure 2F**, we indeed analyzed the immunophenotype in the three conditions (NT, *EVI1* sh#16, and *EVI1* sh#87). Although this experiment was not the primary focus of our investigation and not included in the manuscript, we made an interesting observation that CD64 (*FCGR1A*) was overexpressed in the knockdown cells, and confirmed by flow cytometry. This finding was consistent with the single-cell RNA sequencing (scRNA) experiment presented in **Figure 5** where CD64 increases upon treatment. While the identification of CD64 is intriguing, a comprehensive characterization of the role of this protein would extend beyond the scope of the current manuscript. Therefore, we have decided to reserve the idea for further investigation in a future study, where we can delve deeper into the role of CD64 in the context of *EVI1* regulation.

Page 16 line 318: I think the authors meant “HDACIs” instead of “HDAC”.

We apologize for the mistake.

The authors hypothesized that HDACIs can impair the function of the *EVI1* co-transcriptional complex and profiled by RNAseq two AML cell lines with high expression of *EVI1* post 16 hours treatment with AR-42 and entinostat and found a strong association with *MYC* signature genes. One key experiment that is missing to demonstrate the deregulation of the *EVI1* transcriptional complex is the analysis of *EVI1* chromatin-binding together with H3K27ac profiling by ChIP-seq (or CUT&RUN) in cells pre and post treatment.

We have performed the experiment suggested by the Reviewer to analyze the deregulation of the *EVI1* transcriptional complex. We conducted *EVI1* chromatin-binding analysis along with H3K27ac profiling using ChIP-seq in HNT34 cell line before and after treatment with AR-42 0.8 μ M and entinostat 4 μ M. This experiment is presented in **Figure R5A** (or **Figure S7H** of the new version of the manuscript).

Analysis of the ChIP-Seq data revealed a quadruple *EVI1* binding sites located 1.0 Mb 3' of the *Myc* promoter within a region with high levels of H3K27, a histone mark associated with active transcriptions. The first two peaks cover a region of 1 Kb proximal to the *MYC* first exon, aligning with distribution pattern compatible with the P0, P1 and P2 *MYC* promoter regions³². A third peak identifies a single discrete peak mapping to the 3' region of the first *MYC* intron, coinciding with the mapping of *MYC* P3 promoter region, an alternative starting site^{33, 34}. The last peak is closer to the 3' region of exon 2 of the *MYC* gene. Furthermore, *EVI1* binds the promoter regions of all the three genes *PA2G4*, *FBL*, *DDX21* identified from the intersection of RIME and the transcriptional analysis of HDACis treated 3q26 AML cells (**Figure 6C**). Importantly, HDACis remove *EVI1* and H3K27 binding from these regions suggesting the requirement of *EVI1* for the modulation of a *Myc* signaling related genes in 3q26 models. These findings not only validate our working hypothesis but also provide an explanation for the data presented in the manuscript.

The scRNAseq analysis on follow-up samples provides interesting findings on how the combination azacitidine and entinostat targets stem and progenitor cells, however, as the

authors are aware, it does not distinguish the effect of the single drugs. Can the authors perform cell viability assay and single cell RNAseq on normal CD34+ overexpressing EVI1 (for example by viral transduction) or in PDX cells before and after treatment with the combination and single azacitidine and entinostat?

We have conducted the suggested experiment using patient-derived leukemia xenograft (PDX) cells derived from PDLX_PR#008. In this study, we evaluated the effects of different treatments, including a vehicle control, azacitidine at 1 mg/kg, entinostat at 10 mg/kg, and the combination of both. Our results demonstrate that entinostat effectively suppresses 3q26 leukemic infiltration (**Figure R5B-C** and **R5F-G** that we included in the manuscript as **Figure 5G-H** and **5K-L**) and Myc signaling (**Figure R5D-E** or **5I-J** of the new version of the manuscript). The addition of the hypomethylating agent (azacitidine) did not result in an additional desired synergistic effect.

The findings are further supported by the pharmacodynamic analysis, as depicted in **Figure S6E**, which reveals the suppression of EVI1 expression in AML upon entinostat treatment but not with azacitidine. This suggests that in the context of our study, the combined treatment did not exhibit an augmented effect on EVI1 suppression beyond what was achieved with entinostat alone. These results provide valuable insights into the potential limitations of hypomethylating agents in 3q26 AML.

REFERENCES

1. Finlay-Schultz J, *et al.* Breast Cancer Suppression by Progesterone Receptors Is Mediated by Their Modulation of Estrogen Receptors and RNA Polymerase III. *Cancer Res* **77**, 4934-4946 (2017).
2. Heslop JA, Pournasr B, Duncan SA. Chromatin remodeling is restricted by transient GATA6 binding during iPSC differentiation to definitive endoderm. *iScience* **25**, 104300 (2022).
3. Heslop JA, Pournasr B, Liu JT, Duncan SA. GATA6 defines endoderm fate by controlling chromatin accessibility during differentiation of human-induced pluripotent stem cells. *Cell Rep* **35**, 109145 (2021).
4. Legrand N, *et al.* PPAR β/δ recruits NCOR and regulates transcription reinitiation of ANGPTL4. *Nucleic Acids Res* **47**, 9573-9591 (2019).
5. Li Y, *et al.* A mutant form of ER α associated with estrogen insensitivity affects the coupling between ligand binding and coactivator recruitment. *Sci Signal* **13**, (2020).
6. Mohammed H, *et al.* Endogenous purification reveals GREB1 as a key estrogen receptor regulatory factor. *Cell Rep* **3**, 342-349 (2013).
7. Mohammed H, Taylor C, Brown GD, Papachristou EK, Carroll JS, D'Santos CS. Rapid immunoprecipitation mass spectrometry of endogenous proteins (RIME) for analysis of chromatin complexes. *Nat Protoc* **11**, 316-326 (2016).
8. Serandour AA, Mohammed H, Miremadi A, Mulder KW, Carroll JS. TRPS1 regulates oestrogen receptor binding and histone acetylation at enhancers. *Oncogene* **37**, 5281-5291 (2018).

9. Sottnik JL, *et al.* Mediator of DNA Damage Checkpoint 1 (MDC1) Is a Novel Estrogen Receptor Coregulator in Invasive Lobular Carcinoma of the Breast. *Mol Cancer Res* **19**, 1270-1282 (2021).
10. Venturutti L, *et al.* TBL1XR1 Mutations Drive Extranodal Lymphoma by Inducing a Pro-tumorigenic Memory Fate. *Cell* **182**, 297-316.e227 (2020).
11. Mohammed H, *et al.* Progesterone receptor modulates ER α action in breast cancer. *Nature* **523**, 313-317 (2015).
12. Witwicki RM, *et al.* TRPS1 Is a Lineage-Specific Transcriptional Dependency in Breast Cancer. *Cell Rep* **25**, 1255-1267.e1255 (2018).
13. Dufour CR, *et al.* The mTOR chromatin-bound interactome in prostate cancer. *Cell Rep* **38**, 110534 (2022).
14. Bard-Chapeau EA, *et al.* EVI1 oncoprotein interacts with a large and complex network of proteins and integrates signals through protein phosphorylation. *Proc Natl Acad Sci U S A* **110**, E2885-2894 (2013).
15. Paredes R, *et al.* EVI1 oncoprotein expression and CtBP1-association oscillate through the cell cycle. *Mol Biol Rep* **47**, 8293-8300 (2020).
16. Petruccelli LA, Pettersson F, Del Rincón SV, Guilbert C, Licht JD, Miller WH, Jr. Expression of leukemia-associated fusion proteins increases sensitivity to histone deacetylase inhibitor-induced DNA damage and apoptosis. *Mol Cancer Ther* **12**, 1591-1604 (2013).
17. Odenike OM, *et al.* Histone deacetylase inhibitor romidepsin has differential activity in core binding factor acute myeloid leukemia. *Clin Cancer Res* **14**, 7095-7101 (2008).
18. Lin RJ, Sternsdorf T, Tini M, Evans RM. Transcriptional regulation in acute promyelocytic leukemia. *Oncogene* **20**, 7204-7215 (2001).
19. Valiuliene G, Stirblyte I, Cicenaitė D, Kaupinis A, Valius M, Navakauskiene R. Belinostat, a potent HDACi, exerts antileukaemic effect in human acute promyelocytic leukaemia cells via chromatin remodelling. *J Cell Mol Med* **19**, 1742-1755 (2015).
20. Nishioka C, Ikezoe T, Yang J, Takeuchi S, Koeffler HP, Yokoyama A. MS-275, a novel histone deacetylase inhibitor with selectivity against HDAC1, induces degradation of FLT3 via inhibition of chaperone function of heat shock protein 90 in AML cells. *Leuk Res* **32**, 1382-1392 (2008).
21. Nebbioso A, *et al.* c-Myc Modulation and Acetylation Is a Key HDAC Inhibitor Target in Cancer. *Clin Cancer Res* **23**, 2542-2555 (2017).
22. Gojo I, *et al.* Phase I and pharmacologic study of MS-275, a histone deacetylase inhibitor, in adults with refractory and relapsed acute leukemias. *Blood* **109**, 2781-2790 (2007).

23. Prebet T, *et al.* Prolonged administration of azacitidine with or without entinostat for myelodysplastic syndrome and acute myeloid leukemia with myelodysplasia-related changes: results of the US Leukemia Intergroup trial E1905. *J Clin Oncol* **32**, 1242-1248 (2014).
24. Minucci S, Pelicci PG. Histone deacetylase inhibitors and the promise of epigenetic (and more) treatments for cancer. *Nat Rev Cancer* **6**, 38-51 (2006).
25. Fallahi-Sichani M, Honarnejad S, Heiser LM, Gray JW, Sorger PK. Metrics other than potency reveal systematic variation in responses to cancer drugs. *Nat Chem Biol* **9**, 708-714 (2013).
26. Jang IS, Neto EC, Guinney J, Friend SH, Margolin AA. Systematic assessment of analytical methods for drug sensitivity prediction from cancer cell line data. *Pac Symp Biocomput*, 63-74 (2014).
27. Subramanian A, *et al.* A Next Generation Connectivity Map: L1000 Platform and the First 1,000,000 Profiles. *Cell* **171**, 1437-1452.e1417 (2017).
28. Efron B. Bootstrap methods: another look at the jackknife. In: *Breakthroughs in statistics: Methodology and distribution*. Springer (1992).
29. Coombes BJ, Biernacka JM. Application of the parametric bootstrap for gene-set analysis of gene-environment interactions. *Eur J Hum Genet* **26**, 1679-1686 (2018).
30. Matsuo Y, Adachi T, Tsubota T, Imanishi J, Minowada J. Establishment and characterization of a novel megakaryoblastic cell line, MOLM-1, from a patient with chronic myelogenous leukemia. *Hum Cell* **4**, 261-264 (1991).
31. Gröschel S, *et al.* A single oncogenic enhancer rearrangement causes concomitant EVI1 and GATA2 deregulation in leukemia. *Cell* **157**, 369-381 (2014).
32. Pullner A, Mautner J, Albert T, Eick D. Nucleosomal structure of active and inactive c-myc genes. *J Biol Chem* **271**, 31452-31457 (1996).
33. Levens D. Disentangling the MYC web. *Proc Natl Acad Sci U S A* **99**, 5757-5759 (2002).
34. Nanbru C, *et al.* Alternative translation of the proto-oncogene c-myc by an internal ribosome entry site. *J Biol Chem* **272**, 32061-32066 (1997).

Figure R1. (A) Venn diagram showing the overlap of proteins identified by RIME in EVI1 and IgG samples for each technical (R1 and R2) and biological (HNT34, UCSD/AML1, and PR#003) replicates. (B) Pearson correlation analysis of LFQ intensities. Scatter plot showing the correlation of LFQ signals between technical (R1 and R2, top panel) and biological (PR#003, UCSD/AML1, and HNT34, bottom panel) replicates of the RIME experimental setting. Protein hits with a fold change greater than 4 and a *p*-value less than 0.01 are shown in dark yellow. In the left panel, blue-labeled proteins are the confirmed MYC-related factors identified by intersecting RIME and ssGSEA data, as described in the manuscript. In the right panel, red-labeled proteins are EVI1 interactors identified using isotope labeling-based MS, as described in the supporting literature. (D) Biochemical validation of EVI1 and PA2G4 interaction. Western blot showing EVI1 and PA2G4 in protein lysates extracted from HNT34 cells (top) or 293T cells (bottom) transduced with a Lenti-hEF1a-ORF-P2A-eGFP-IRES-Puro vector (Transomic, Huntsville, AL, USA, #TLO2015.1) containing the *EV11* (NM_005241.3) and the *PA2G4* (GeneBankBC001951.1) ORFs. Cell lysates were immunoprecipitated with an anti-EVI1 or anti-PA2G4 antibody. An anti-IgG antibody was used as a control of immunoprecipitation. The anti-EVI1 (Cell Signaling Technology, Danvers, MA, USA, C50E12, #2593) and the anti-IgG (Rabbit, Merk Darmstadt, Germany, #12-370) antibodies were the same used in RIME.

Figure R2. (A) EVI1, ΔEVI1 expression level in HNT34 and HL-60 cells transduced with an empty or an ORF-EVI1 cdna vector. β-Actin was used as loading control. (B) Representative IF showing HL-60 cells transduced with an empty (top panels) or an ORF-EVI1 cdna (bottom panels) vector and stained with an anti-EVI1 antibody (in red). Nuclei in blue (DAPI). Scale bar: 100 μm. (C) Effect of HDACis and ara-C on cell viability after 72 hr of treatment in HL-60 ± ORF-EVI1 cdna. (D) Expression of EVI1 in MOLM1 sorted based on cell size. Large cells express EVI1 compared to small cells. (E) Abnormal 3q26 pattern on fluorescence in situ hybridization (FISH) in MOLM1 sorted by cell size. The break-apart hybridization pattern 1F1G1O (one fusion and two separated signals, one green and one orange) indicates the break and split of the EVI1 locus. The abnormal pattern was observed in cells with large nuclei. At least 100 nuclei/cells were analyzed. (F) Effects of HDACis and chemotherapy treatment on viability in MOLM1 EVI1^{high} and MOLM1 EVI1^{low} after 72 hr of drug exposure. (G) EVI1, ΔEVI1 and MYC expression level HNT34, U937T (clone E10) +/- tetracycline and MYC overexpressing U937T cells. β-Actin was used as loading control. (H) Effects of AR-42, belinostat, and entinostat on viability in U937T +/- tetracycline and MYC overexpressing U937T cell lines following 72h of drug treatment at the indicated concentrations. Statistical significance (*P ≤ 0.05, **P ≤ 0.01, ***P ≤ 0.001, ****P ≤ 0.0001) was determined by a multiple unpaired t-test (C), one-way ANOVA with Sidak's correction for multiple comparison testing (F) or one-way ANOVA using Tukey's correction for multiple comparison testing (H). Data are presented as mean ± SD in C (n=2), F (n=2) and H (n = 2) .

Figure R3. (A) Western blot analysis showing the expression of EVI1 and Δ EVI1 in HNT34 cells six days after shRNA transduction. **(B)** Percentage of EVI1 mRNA relative to the control gene *RPL13A* ($\Delta\Delta$ CT) in HNT34 cells six days after shRNA transduction. **(C)** Effect of EVI1 loss in HNT34 cells at three, six, or nine days after shRNA selection. **(D)** BRAID index analysis for the combinations of WS6 with HDACi (AR-42, belinostat and entinostat) in 2 EVI1^{High} (samples PR#024 - PDLX#PR#008) and 3 EVI1^{Low} (samples PR#034 - PR#031 - PR#039) primary AML samples treated for 72h. A color scale bar represents the level of drug antagonism or synergism. K index is indicated. A positive K index indicates a synergistic effect. **(E)** EVI1, Δ EVI1 and MYC expression level in a panel of EVI1^{High} (left) and EVI1^{Low} (right) AML cell lines. β -Actin was used as loading control. **(F)** Effect of AR-42, belinostat and entinostat on 6 MYC^{High} (OCI/AML2, OCI/AML3, IMS-M2, GDM1, NOMO1 and TF1) and 6 MYC^{Low} (SKM1, MOLM13, MOLM1, UCSD/AML1, HNT34 and MUTZ-3) AML cell lines. The AUC model of the log-transformed dose-response data is depicted. **(G)** Effect of JQ-1 on 6 MYC^{High} (OCI/AML2, OCI/AML3, IMS-M2, GDM1, NOMO1 and TF1) and 6 MYC^{Low} (SKM1, MOLM13, MOLM1, UCSD/AML1, HNT34 and MUTZ-3) AML cell lines. The AUC model of the log-transformed dose-response data is depicted. **(H)** Effect of WS6 in 8 EVI1^{High} (samples PR#002 - 006, PR#008 and PR#023 - 024) and 15 EVI1^{Low} (samples PR#025-039) primary AML samples. The AUC model of the log-transformed dose-response data is depicted. **(I)** Representative metaphase spread (left) and karyotyping (right) of MOLM1 cells (Q-banding). Statistical significance (* $P \leq 0.05$, ** $P \leq 0.01$, **** $P \leq 0.0001$) was determined by one-way (B) or two-way (C) ANOVA using Tukey's correction for multiple comparison testing and a non-parametric t-test (Mann-Whitney) (F, G and H). Data are presented as mean \pm SD in B (n=3), C (n=5), F-G (MYC^{High}=6, MYC^{Low}=6) and H (n \geq 8).

Figure R4. (A) Volcano plot of RNaseq expression in $EVI1^{High}$ or $EVI1^{Low}$ AML cell lines segregated based on the 3q26 status derived from the E-MTAB-2225 dataset. DEG are depicted in violet if repressed (\log_2 fold change ≤ -2 , $Adj.P \leq 0.01$) or in dark yellow if upregulated (\log_2 fold change $\geq +2$, $Adj.P \leq 0.01$). (B) Volcano plot derived from RNaseq gene expression data of HNT34 cells transduced with a non-targeting shRNA (Control) or after $EVI1$ -directed shRNAs (sh#16 and sh#87) three days after selection (see also Figure S3A). DEG are depicted in violet if upregulated in Control (\log_2 fold change ≤ -2 , $Adj.P \leq 0.05$) or in dark yellow if upregulated in $EVI1$ shRNA (\log_2 fold change $\geq +2$, $Adj.P \leq 0.05$). (C) GSEA analysis of Affymetrix data from Figure S2A using the oncogenic gene sets from HNT34 transduced with shRNAs targeting $EVI1$. Enrichment plot of the $EVI1$ overlapping genes is shown. NES, normalized enrichment score; FDR, false discovery rate. (D) L1000FWD fireworks visualization of drug-induced signatures mimicking and reversing the differential gene expression signature generated from $EVI1^{High}$ (n=5) or $EVI1^{Low}$ (n=2) AML cell lines contained in the E-MTAB-2225 dataset (left) and $EVI1$ -silenced HNT34 (right). Aquamarine circles indicate small molecule causing a reverse signature $EVI1$ "Off". Light violet circles indicate small molecule mimicking a signature $EVI1$ "On". HDACis drug-induced signatures are indicated in green overlapping with aquamarine circles. (E) Bar plot displaying top drugs inducing an $EVI1$ "Off" status identified by the L1000CDS2 query of E-MTAB-2225 (from Figure R4A, left) and HNT34 (from Figure R4B, right) datasets. HDACi are highlighted in green. (F) Violin plots comparing ΔPOC activity between HDACis and the indicated drug classes. "n" indicates number of small molecules within each class. Statistical significance among groups (*** $P \leq 0.001$, **** $P \leq 0.0001$) was determined by one-way ANOVA (F) using Tukey's correction for multiple comparison testing.

Figure R5. (A) Tracks showing EVI1 binding and H3K27ac enrichment across the *MYC* locus in HNT34 cells treated with DMSO, AR-42 and entinostat. The bottom bar represents the genes (hg19), and the y-axis represents normalized read density scaled to 1 million. On the right panel real-time PCR after ChIP performed in HNT34 cells treated with DMSO, AR-42 and entinostat at the indicated doses using an anti-EVI1 and H3K27Ac antibody and primers targeting the *MYC* promoter. Results are expressed as fold enrichment of *EVI1* and *H3K27Ac* compared to a non-specific IgG antibody. **(B)** *In vivo* antileukemic effect of entinostat and azacitidine. Mice (PDLX_PR#008) were treated with vehicle (DMSO), azacitidine (1 mg/kg) for 5 days, entinostat (10 mg/kg) (5 days/week) or the combination of both for 3 weeks. Histograms show the percentage of CD45⁺ in bone marrow (left) and the percentage of *EVI1* mRNA relative to the control gene *RPL13A* ($\Delta\Delta CT$) in CD45⁺ cells (right) at the end of treatment. **(C)** UMAP plot of clustering results of PDLX_PR#008 bone marrow LP before (i) and after entinostat (ii), azacitidine (iii) or the combination of both (iv), colored according to the UCell score of leukemic markers. **(D)** GSEA running score plot of the top enriched *MYC* targets V1 pathway in PDLX_PR#008 LP after the treatment with azacitidine/entinostat ($Adj.P = 3.40 \times 10^{-8}$) or entinostat as single agent ($Adj.P = 2.17 \times 10^{-12}$). Each graph indicates the running enrichment score (ES) of the pathway (top), the location of single genes of the gene set in the ranking (central), and the distribution of the ranking metric (bottom). **(E)** Dot plot of GSEA results illustrating Molecular Signatures Database (MsigDB) biological processes associated with the indicated treatments in comparison to the vehicle in PDLX_PR#008 LP. Set size refers to the number of genes associated with each (MsigDB) biological process. Dot color indicates the range of (BH) Adj.P for each pathway. **(F)** EVI1, MYC, and Ki67 expression (brownish) in bone marrow leukemic cells (PDLX_PR#008) following three weeks treatment of azacitidine and entinostat as described in panel G. FFPE tissue were stained with anti-EVI1, anti-MYC and anti-Ki67 antibodies revealed by immunoperoxidase. Scale bar: 100 μm . **(G)** Histograms display the mean \pm SD of the percentage of the EVI1, MYC or Ki67-positive cells in FFPE tissue sections. Statistical significance ($*P \leq 0.05$, $**P \leq 0.01$, $***P \leq 0.001$, $****P \leq 0.0001$) was determined by one-way ANOVA using Tukey's correction for multiple comparison testing. Data are presented as mean \pm SD in A (n = 3), B (vehicle=7, treated=5 for the left panel and n=9 for the right panel) and G (n=10 fields per condition).

Reviewers' Comments:

Reviewer #1:

Remarks to the Author:

The authors have adequately addressed the issues raised by this reviewer.

Reviewer #2:

Remarks to the Author:

The authors have addressed my comments satisfactorily.

Reviewer #3:

Remarks to the Author:

The authors made substantial efforts in addressing the comments raised by the Reviewers. The additional experiments and analyses included in this revised version enhanced the quality of the manuscript. Based on these findings and experimental clarifications, I do not have further comments.

We would like to thank the Reviewers for their work on this manuscript and for recognizing our efforts to address the scientific points and explanations raised in their revisions.

Reviewer #1:

The authors have adequately addressed the issues raised by this reviewer.

The authors thank this reviewer for their insightful technical revision of the RIME assay. Their suggested analysis and raised clarifications helped us significantly improve the clarity of this study section.

Reviewer 2:

The authors have addressed my comments satisfactorily.

We thank this reviewer for the suggested approach and the clarification they provided. This helped to make our manuscript more robust and readable.

Reviewer #3:

The authors made substantial efforts in addressing the comments raised by the Reviewers. The additional experiments and analyses included in this revised version enhanced the quality of the manuscript. Based on these findings and experimental clarifications, I do not have further comments.

The authors thank this reviewer for the suggested experiments and explanations that strengthened our manuscript. We also appreciate the kind words.